

# The high frequency response correction of eddy covariance fluxes. Part 2: the empirical approach and its interdependence with the time-lag estimation

Olli Peltola[1], Toprak Aslan[2], Andreas Ibrom[3], Eiko Nemitz[4], Üllar Rannik[2], and Ivan Mammarella[2]

[1]Climate Research Programme, Finnish Meteorological Institute, P.O. Box 503, 00101 Helsinki, Finland
[2]Institute for Atmospheric and Earth System Research (INAR)/Physics, Faculty of Science, University of Helsinki, P.O. Box 68, 00014 Helsinki, Finland
[3]Dept. Environmental Engineering, Technical University of Denmark (DTU), Lyngby, Denmark
[4]UK Centre for Ecology and Hydrology (UKCEH), Edinburgh Research Station, Penicuik, Bush Estate, EH26 0QB, UK

**Correspondence:** Olli Peltola (olli.peltola@fmi.fi)

**Abstract.** The eddy covariance (EC) technique has emerged as the prevailing method to observe ecosystem - atmosphere exchange of gases, heat and momentum. EC measurements require rigorous data processing to derive the fluxes that can be used to analyse exchange processes at the ecosystem - atmosphere interface. Here we show that two common post-processing steps (time-lag estimation via cross-covariance maximisation, and correction for limited frequency response of the EC measurement system) are interrelated and this should be accounted for when processing EC gas flux data. These findings are applicable to EC systems employing closed- or enclosed-path gas analysers which can be approximated to be linear first-order sensors. These EC measurement systems act as a low-pass filters on the time-series of the scalar $\chi$ (e.g. $CO_2$, $H_2O$) and this induces a time-lag ($t_{lpf}$) between vertical wind speed ($w$) and scalar $\chi$ time series which is additional to the travel time of the gas signal in the sampling line (tube, filters). Time-lag estimation via cross-covariance maximisation inadvertently accounts also for $t_{lpf}$ and hence overestimates the travel time in the sampling line. This results in a phase shift between the time-series of $w$ and $\chi$, which distorts the measured cospectra between $w$ and $\chi$ and hence has an effect on the correction for dampening of EC flux signal at high frequencies. This distortion can be described with a transfer function related to the phase shift ($H_p$) which is typically neglected when processing EC flux data. Based on analyses using EC data from two contrasting measurement sites, we show that the low-pass filtering induced time-lag increases approximately linearly with the time constant of the low-pass filter, and hence the importance of $H_p$ in describing the high frequency flux loss increases as well. Incomplete description of these processes in EC data processing algorithms results in flux biases of up to 10%, with the largest biases observed for short towers due to prevalence of small scale turbulence. Based on these findings, it is suggested that spectral correction methods implemented in EC data processing algorithms are revised to account for the influence of low-pass filtering induced time-lag.

## 1 Introduction

The eddy covariance (EC) is the standard micrometeorological technique for measuring vertical turbulent fluxes of momentum, heat and gases in the atmospheric surface layer (Aubinet et al., 2012). Under certain conditions (flat terrain, homogeneous and





stationarity turbulent flows, source/sink homogeneity, absence of chemical sources/sinks), the measured EC fluxes represent direct and continuous estimates of the surface exchange of energy and matter between the surface and the atmosphere, and are then widely used to estimate energy and water balances, as well as carbon budget of different type of ecosystems (Baldocchi, 2003; Vesala et al., 2008; Mammarella et al., 2015).

Gas flux measurements are made using a three-dimensional sonic anemometer and a gas analyser, which are able to provide fast-response measurements of turbulent fluctuations of vertical wind velocity and gas concentration (Aubinet et al., 2012). All EC systems, including open-path, enclosed-path or closed-path gas analysers, act as low pass filters, and thus cause a systematic bias to flux estimates. Flux loss at high frequency is due to the incapability of the measurement system to detect small-scale variation. The main reasons for co-spectral attenuation are inadequate frequency response of the sensor, sensor separation and

line averaging, as well as, in closed-path systems, the sampling of air trough tubes and filters.

    The frequency response correction is usually performed based on *a priori* knowledge of the system transfer function and the unattenuated cospectrum:

$$CF = \frac{\int_{f_1}^{f_2} Co^{w,\chi}\, df}{\int_{f_1}^{f_2} Co^{w,\chi} H_{lpf}\, df}. \tag{1}$$

    Here CF is the estimated spectral correction factor, $Co^{w,\chi}$ the normalised unattenuated cospectrum between scalar ($\chi$) and

vertical wind speed ($w$), $f$ the frequency, $H_{lpf}$ the total transfer function describing the low-pass filtering, and $f_1$ and $f_2$ are the integration limits that are determined by the length of the averaging period and Nyquist frequency, respectively. The correction, performed by multiplying the measured covariance by the factor CF, always increases the absolute flux magnitude. Note that the low-frequency transfer function (Rannik and Vesala, 1999) is not included in Eq. (1), as here we focus only on the low-pass filtering effect of the EC system. The high-frequency transfer function $H_{lpf}$ can be derived either theoretically or empirically

(Foken et al., 2012a). The correction is in general different for momentum flux, sensible and latent heat fluxes and the various gas fluxes, and is specific to each EC system. In the theoretical approach, the atmospheric surface layer co-spectral models (Moncrieff et al., 1997) are used, and $H_{lpf}$ is calculated as a convolution of specific transfer functions representing different causes of flux loss, the equations for which can be found in Moncrieff et al. (1997) and Moore (1986). This approach works well for correcting the fluxes of momentum and sensible heat, as well as gas fluxes measured by open-path systems. Further,

Horst (1997) and Massman (2000) have proposed alternative theoretical approaches, providing an analytical estimation of correction factors.

    Alternatively, the empirical approach can be used, where the model cospectra and $H_{lpf}$ are estimated using in situ measurements. This is the preferable approach for closed-path systems, where additional effects related to the inlet dust filters and rain caps (Aubinet et al., 2016; Metzger et al., 2016), potential variations in the sampling line flow rate, and absortion/desorption

processes inside the sampling tube (Ibrom et al., 2007a; Nordbo et al., 2014; Runkle et al., 2012) may contributed to the EC system cut-off frequency, introducing transfer functions which are difficult to estimate *a priori*.

    For this approach, different methods have been proposed for retrieving $H_{lpf}$ from the measured power spectra or cospectra of the sonic temperature $T$ and the target gas dry mole fraction (Fratini et al., 2012; Ibrom et al., 2007a; Mammarella et al.,





2009; Nordbo et al., 2014). While the use of power spectra is the method recommended by the ICOS methodology (Sabbatini et al., 2018; Nemitz et al., 2018) and is implemented in the EddyPro software package, other studies and software packages (EddyUH) have supported the use of measured cospectra (Aubinet et al., 2000; Mammarella et al., 2009, 2016) for the empirical retrieval of $H_{lpf}$. The companion paper (Aslan et al., 2020) has investigated methodological issues and flux uncertainty related

to these two methods under different attenuation conditions and signal-to-noise ratio scenarios.

In EC systems, particularly those using closed-path gas analysers, the measurements of vertical wind velocity and gas concentration are not co-located, and a time delay between the two signals exists (Aubinet et al., 2012). The standard procedure is to determine this time delay for each averaging period (typically 30 minutes) by maximising the related cross-covariance function (in absolute terms), and taking the corresponding lag as the true signal delay. However, this time shift between the two

signals depends not only on the system setup (e.g. air sample travel time from the inlet to the IRGA sampling cell, separation distance between the inlet and the centre of the sonic anemometer path), but it can additionally reflect the low-pass filtering of the measured signal (Massman, 2000; Ibrom et al., 2007b, a).

In this study, we investigate how the low-pass filtering induced phase shift affects the estimation of the high frequency flux loss, and we show the implications that occur when $H_{lpf}$ time constants are empirically derived from measured power spectra

or cospectra, and related CFs values are applied to covariances estimated from the cross-covariance maximum. Finally, we present a method to take the phase shift effect into account when processing EC data. Towards these aims, we use EC data collected above a forest canopy (Hyytiälä) and a wetland (Siikaneva) in Finland covering a large range of attenuation conditions and integral turbulent time scale characteristics.

## 2   Theory

### 2.1   Transfer functions

The measured cross-spectrum ($Cr_m$) between the fluctuations of vertical wind speed ($w'$) and the attenuated and lagged scalar ($\hat{\chi}'$; in this paper attenuated variables are denoted with ˆ) can be described by (e.g. Massman (2000)):

$$Cr_m = [Z_w][h_\chi Z_\chi]^* e^{-j\phi_{phys}}, \tag{2}$$

where $Z_w$ and $Z_\chi$ are Fourier transforms of the time series of $w'$ and $\chi'$ which are perfectly in phase and not attenuated, $h_\chi$ is the

Fourier transform of the filter function that describes the response of the instrument measuring the scalar time series, $j = \sqrt{-1}$ and $\phi_{phys} = \omega t_{phys}$, where $t_{phys}$ is a constant time shift (s) between $w'$ and $\chi'$ and $\omega = 2\pi f$ is the angular frequency, with $f$ representing the frequency (Hz). The superscript $^*$ denotes complex conjugation. Equation (2) describes the cross-spectra measured with a typical EC system employing closed-path gas analyser where the scalar fluctuations are attenuated ($h_\chi$), and delayed in time ($t_{phys}$) with respect to $w'$ due to the gas sampling system (tubes, filters) and horizontal sensor separation.

The signal travel time in the gas sampling line ($t_{phys}$) could be approximated from the volume of the sampling line and flow rate. Note that we assume that the sensor measuring $w'$ is perfect without any distortions in the measured $w'$ time series. This





assumption does not limit the generality of the findings below. We also neglect any influence of sensor separation (Horst and Lenschow, 2009) or high-pass filtering on flux attenuation.

Following, e.g., Horst (1997) and Massman (2000), the scalar sensor is approximated to be a linear first-order sensor for which $h_\chi$ can be written as

$$h_\chi = \frac{1}{1 - j\omega\tau}, \tag{3}$$

where $\tau$ is the time constant (s) of the filter, also called response time. Note that $Z_w Z_\chi^*$ is the true cross-spectrum ($Cr$) between unattenuated $w'$ and $\chi'$ time series which are in phase and it has a real (cospectrum $Co$) and an imaginary (quadrature spectrum $Q$) part ($Cr = Z_w Z_\chi^* = Co + jQ$). Recall also that the integral of $Co$ equals the unattenuated covariance $\overline{w'\chi'}$, i.e. the vertical turbulent flux of scalar $\chi$. Following previous studies (Horst, 1997; Massman, 2000), the quadrature spectrum ($Q$) can be assumed to be zero for stationary turbulent flow and hence $Cr = Z_w Z_\chi^* \approx Co$. Now, using Euler's formula and some derivation (see Appendix A), we find

$$Cr_m = \frac{\cos\phi_{phys} - \omega\tau\sin\phi_{phys}}{1 + \omega^2\tau^2} Co - j\frac{\sin\phi_{phys} + \omega\tau\cos\phi_{phys}}{1 + \omega^2\tau^2} Co = Co_m + jQ_m, \tag{4}$$

where the first term on the right equals the measured cospectrum ($Co_m$) and the second term the measured quadrature spectrum ($Q_m$). Note that whilst $Q$ was assumed to be negligible, $Q_m$ is in fact not equivalent to zero even when $\phi_{phys} = 0$, i.e. when the travel time of scalar $\chi'$ signal in the sampling system is zero. This means that $w'$ and the attenuated $\hat{\chi}'$ (ˆ denotes attenuation throughout the text), depending on frequency, can be out of phase due to low-pass filtering of $Z_\chi$ with $h_\chi$. This has already previously been noted (e.g. Massman, 2000; Massman and Lee, 2002; Wintjen et al., 2020). Now, simply based on the real part of $Cr_m$, we can define a transfer function for the first-order system ($H$) as

$$H = \frac{1}{1 + \omega^2\tau^2}, \tag{5}$$

and a transfer function ($H_p$) for a generic phase shift $\phi$ as (Massman, 2000):

$$H_p = \cos\phi - \omega\tau\sin\phi. \tag{6}$$

These two transfer functions together describe how the measured cospectrum ($Co_m$) deviates from the cospectrum calculated from ideal measurements free from any attenuation and time shifts ($Co$).

Similarly, forming the cross-spectrum of the attenuated scalar with itself yields:

$$Cr_{m,p} = [h_\chi Z_\chi][h_\chi Z_\chi]^* \tag{7}$$

$$= \frac{1}{1 + \omega^2\tau^2} Z_\chi Z_\chi^* = H Z_\chi Z_\chi^*, \tag{8}$$

where $Z_\chi Z_\chi^*$ is the unattenuated power spectrum. From this it follows that the same transfer function (i.e. $H$) applies to both power spectrum and cospectrum in the case that there is no phase shift between $w'$ and $\chi'$ (i.e. $H_p$ equals 1) and the quadrature spectrum $Q$ is zero.





## 2.2 Time lag determination via cross-covariance maximisation

The widely used method to estimate the time lag between $w'$ and $\hat{\chi}'$ via cross-covariance maximisation can be considered to be equivalent to finding a time shift $t$ (and hence $\phi = \omega t$) that maximises the integral of $Co_m$ above (i.e. maximises the covariance between $w'$ and $\hat{\chi}'$):

$$\int_0^\infty \frac{\cos(\omega t) - \omega \tau \sin(\omega t)}{1 + \omega^2 \tau^2} Co\, d\omega = \overline{w' \hat{\chi}'}(t) \tag{9}$$

The original aim of this cross-covariance maximisation is to account for the time lag between the time series $w'$ and $\hat{\chi}'$ that is induced by sampling lines (air filters, tubing) and horizontal sensor separation, in other words to find such time lag that removes the $e^{-j\phi_{phys}}$ term in Eq. (2). After such successfully accounting for the travel time ($t_{phys}$), the attenuation of the measured cospectrum $Co_m$ (and hence flux) would be described only by $H$, instead of $H$ and $H_p$ (see Eq. (4)), since the time lag estimation sets $H_p = 1$. However, the filter described by Eq. 3 results in a phase shift and hence produces an additional time lag between $w'$ and $\hat{\chi}'$, i.e. $t_{lpf}$. This can be seen from Eq. (4): $Q_m$ differs from zero when $\phi = 0$ (see also Massman (2000); Massman and Lee (2002); Wintjen et al. (2020)). Cross-covariance maximisation inadvertently derives the sum of $t_{phys}$ and $t_{lpf}$, which implies that the time lag determined by cross-covariance maximisation is not the desired transport time lag. Using the sum of $t_{phys}$ and $t_{lpf}$ instead of $t_{phys}$ induces a non-negligible negative $\phi$ in Eq. (4). Hence when shifting the time series according to the cross-covariance maximisation, the transfer function related to the phase shift ($H_p$) can no longer be neglected and should be calculated using $\phi = -\omega t_{lpf}$, where $t_{lpf}$ is the bias in the time lag when estimated via cross-covariance maximisation, in other words the low-pass filtering induced time lag (note the minus-sign, since $H_p$ accounts for the overestimated time lag).

As the maximal possible cospectral energy content per unit frequency can be described with the amplitude spectrum ($A_m = \sqrt{Co_m^2 + Q_m^2}$), it can be assumed that after the cross-covariance maximisation (time series shifted by $t_{phys} + t_{lpf}$), $Co_m$ can be approximated by $A_m$. It is straightforward to derive $A_m$ from Eq. (4):

$$A_m = \sqrt{Co_m^2 + Q_m^2} = \frac{Co}{\sqrt{1 + \omega^2 \tau^2}} = \sqrt{H} Co. \tag{10}$$

Hence $A_m$ is attenuated with $\sqrt{H}$ instead of $H$ which describes the attenuation of $Co_m$ in the case that there is no phase shift between $w'$ and $\hat{\chi}'$ as shown above. This suggests that after cross-covariance maximisation $\sqrt{H}$ approximates $H H_p$ and the correct transfer function for cospectrum calculated after cross-covariance maximisation. Note that Eq. (10) applies universally, independent of $\phi$.

The dependence between low-pass filtering induced time lag ($t_{lpf}$) and filter response time ($\tau$) was estimated using the approximation $H H_p \approx \sqrt{H}$ or $H_p \approx \frac{1}{\sqrt{H}}$. For this analysis the phase transfer function $H_p$ was approximated by using the series expansion of the terms in $H_p$ (Eq. 6). This approximation resulted in $H_p \approx 1 - \frac{1}{2}(\omega t_{lpf})^2 + \omega \tau (\omega t_{lpf})$. Similarly, $\frac{1}{\sqrt{H}}$ can be approximated by $1 + \frac{1}{2}(\omega \tau)^2$. Equating these two approximations up to the second order yields a quadratic equation where the frequency dependence cancels out: $t_{lpf}^2 - 2\tau t_{lpf} + \tau^2 = 0$. The solution of this simple equation gives $t_{lpf} = \tau$. Thus, up to the second order approximation, which holds roughly up to the frequency value $\omega = 1$, the low pass filtering induced time





lag equals the time constant of the filter. Note, however, that at this frequency range ($\omega$ between 0 and 1) the filtering effect is very weak.

## 3 Materials and methods

### 3.1 Measurement sites

Measurements from the Siikaneva fen and the Hyytiälä forest site (SMEAR II) were used. Both stations are part of Integrated Carbon Observation System (ICOS) measurement station network.

The SMEAR II station is situated in Southern Finland (61$^o$51$'$ N, 24$^o$17$'$ E; 181 m a.s.l.). The station is surrounded by extended areas of coniferous forests and the EC tower is located in a 57-year-old (in 2019) Scots pine (*Pinus sylvestris L.*) forest with a dominant tree height of ca. 19 m. The EC measurements were performed with 10 Hz sampling frequency at
27 m height, whereas the zero plane displacement height ($d$) was 14 m. The wind speed components and sonic temperature were measured by a 3-D ultrasonic anemometer (Solent Research HS1199, Gill Ltd, UK), while carbon dioxide and water vapor mixing ratios were measured by an infrared gas analyzer (LI-7200, LI-COR Biosciences, USA). The centre of the sonic anemometer was displaced 20 cm horizontally and 1 cm vertically from the intake of the gas analyzer. The measurement setup was designed to closely follow the ICOS EC measurement protocols (Franz et al., 2018; Rebmann et al., 2018). The
measurements were conducted between May and August 2019.

The Siikaneva fen site is located in Southern Finland (61$^o$49.9610$'$ N, 24$^o$11.5670$'$ E; 160 m a.s.l), consisting mainly of sedges (*Eriophorum vaginatum, Carex rostrata, C. limosa*) and Sphagnum-species, namely *S. balticum, S. majus* and *S. papillosum* with the height of ca. 10–30 cm. Further details about the site can be found in Riutta et al. (2007), Peltola et al. (2013) and Rinne et al. (2018). The EC measurements were conducted between May and August 2013. The data used in
this study were measured with 10 Hz sampling frequency using a 3-D sonic anemometer (Metek USA- 1, GmbH, Elmshorn, Germany) and a closed-path analyzer (LI-7000, LI-COR Biosciences, USA). The sonic anemometer and the gas inlet were situated at 2.75 m above the peat surface, and the air was drawn to the analyzer through a 16.8 m long heated inlet sampling line. The center of the sonic anemometer was displaced 25 cm vertically above the intake of gas analyzer.

A short dataset (hereafter $D_{S_1}$) measured between 9:30 and 11:30 on 16-Jun 2013 at the Siikaneva site was used in the
method validation (see Sect. 4.2), whilst long-term datasets from both Siikaneva (hereafter $D_{S_2}$) and Hyytiälä (hereafter $D_H$) were used to evaluate the accuracy of different spectral correction methods and their effect on $CO_2$ and $H_2O$ fluxes (see Sect. 4.3).

### 3.2 Data processing

High frequency EC data from the two sites were processed in order to evaluate 1) the accuracy of different spectral correction
methods and 2) their effect on gas ($CO_2$ and $H_2O$) flux estimates at these sites. The processing steps followed commonly accepted routines: 1) data were despiked using a method based on the running median filter (Brock, 1986; Starkenburg et al.,





2016), 2) coordinates were rotated using double-rotation which aligned one of the horizontal wind components with the mean wind setting the mean vertical and cross wind components to zero, 3) gas ($CO_2$ and $H_2O$) mole fractions were converted point-by-point to be relative to dry air, if not already done internally by the gas analyser (LI-7200), 4) turbulent fluctuations were extracted from the measurements using block-averaging and 5) time lags between gas or filtered $T$ (see Sect. 3.2.2)

and vertical wind time series were accounted for using cross-covariance maximisation. Power spectra and cospectra were calculated after these processing steps. Spectral corrections and the related correction factors ($CF$) were calculated using four methods described below, see Sect. 3.2.1. After processing, the flux time series were quality filtered by removing periods with unrealistic sensible heat fluxes or friction velocities and highly non-stationary fluxes (Foken and Wichura, 1996). Site specific friction velocity thresholds (0.17 and 0.3 m/s for Siikaneva and Hyytiälä, respectively) were used to discard gas flux data during

low turbulence periods. Furthermore, at Hyytiälä wind directions between $150°$ and $230°$ were discarded since here the mast construction disturbed the air flow. At Siikaneva an omnidirectional sonic anemometer was used mounted to the top of a mast and hence clearly disturbed wind directions were not identified. After quality filtering 4210 and 4722 30-min periods were available for analysis from Hyytiälä and Siikaneva, respectively.

Atmospheric stability was evaluated using the Obukhov length ($L$):

$$L = -\frac{u_*^3}{\frac{v_k g \overline{w'\theta_v'}}{\overline{\theta_v}}}, \tag{11}$$

where $u_*$ is the friction velocity, $v_k = 0.4$ the von Karman constant, $g$ the acceleration due to gravity and $\theta_v$ is virtual potential temperature. Overline denotes temporal averaging and primes deviations from the mean. From the Obukhov length, the stability parameter was calculated as $\zeta = \frac{z-d}{L}$, where $z$ is the measurement height and $d$ is the zero plane displacement height.

### 3.2.1 Empirical estimation of the high-frequency attenuation

The total transfer function of an EC setup can be estimated empirically from measured $w$-$\hat{\chi}$ and $w$-$T$ cospectra:

$$H_{e,cs} = \frac{Co^{w,\hat{\chi}}}{Co^{w,T}} \frac{\overline{w'T'}}{\overline{w'\hat{\chi}'}} F_n, \tag{12}$$

or, alternatively, from the power spectra of the measured scalar and $T$:

$$H_{e,ps} = \frac{S_{\hat{\chi},\hat{\chi}}}{S_{T,T}} \frac{\sigma_T^2}{\sigma_{\hat{\chi}}^2} F_n, \tag{13}$$

where $Co^{w,\hat{\chi}}$ denotes the cospectral density between $w$ and $\hat{\chi}$, and $\overline{w'\hat{\chi}'}$ represents the covariance between $w$ and $\hat{\chi}$. $S_{\hat{\chi},\hat{\chi}}$ and

$\sigma_{\hat{\chi}}^2$ denote the power spectral density and variance of $\hat{\chi}$, respectively, and the corresponding terms are defined for $T$. When estimating $H_{e,cs}$ and $H_{e,ps}$, the covariances and variances were calculated from unattenuated frequencies (between $5 \times 10^{-3}$ Hz and $5 \times 10^{-2}$ Hz) (Foken et al., 2012a; Sabbatini et al., 2018). An additional normalisation factor ($F_n$) was also incorporated in order to take into account the fact that the variances and covariances in Eqs. (12) and (13) may be subject to various high frequency losses (Ibrom et al., 2007a). Here, $w$ and $T$ were approximated to be free from any low-pass filtering effects

and thus $Co^{w,T}$ and $S_{T,T}$ could be used as a reference. $H_{e,cs}$ and $H_{e,ps}$ were calculated from ensemble-averaged spectra





**Table 1.** Methods used in this study to estimate flux losses and related correction factors ($CF$) due to low-pass filtering of the scalar signal. See the definitions for $H$ and $H_p$ in Eqs. (5) and (6), respectively. Values for $\tau$ (and $t_{lpf}$ in method 4)) were estimated with non-linear least squares fit to $H_{e,ps}$ or $H_{e,cs}$ where $F_n$ and $\tau$ (and $t_{lpf}$ in method 4)) were used as fitting parameters. The additional normalisation factor $F_n$ took into account any inaccuracies in estimation of variances and covariances in calculation of $H_{e,ps}$ or $H_{e,cs}$ (Eqs. (13) and (12)), respectively. The fits were weighted with temperature cospectra, in order to give more weight to frequencies where the scalar fluxes were high.

|  | Estimation of $\tau$ | Fitting parameters | $H_{lpf}$ used in the estimation of $CF$, Eq. (1) | Reference |
| --- | --- | --- | --- | --- |
| Method 1 | fit $H$ to $H_{e,ps}$ | $F_n, \tau$ | $H$ | Sabbatini et al. (2018) |
| Method 2 | fit $H$ to $H_{e,cs}$ | $F_n, \tau$ | $H$ | Aubinet et al. (2000) |
| Method 3 | fit $H$ to $H_{e,ps}$ | $F_n, \tau$ | $\sqrt{H}$ | Fratini et al. (2012) |
| Method 4 | fit $HH_p$ to $H_{e,cs}$ | $F_n, \tau, t_{lpf}$ | $HH_p$ | This study |

from measurement periods that fulfilled the following criteria: flux stationarity (Foken and Wichura, 1996) was below 0.3, $u_* > 0.1\,\mathrm{m\,s^{-1}}$, $\overline{w'T'} > 0.02\,\mathrm{K\,m\,s^{-1}}$. For CO$_2$ it was also required that its turbulent flux was below -0.05 $\mu\mathrm{mol\,mol^{-1}\,m\,s^{-1}}$ and for H$_2$O it was required that the flux was directed upwards. For H$_2$O the transfer functions were determined in relative humidity (RH) bins and the $\tau$ and $t_{lpf}$ values used in calculating $CF$ were estimated from exponential fits made to the values

obtained in the RH bins (similarly as for $\tau$ in Mammarella et al. (2009)). In the case of CO$_2$ and H$_2$O fluxes, the power spectra were subjected to noise removal prior utilising them in Eq. (13) by assuming that the signal was contaminated by white noise and that at the highest frequencies the power spectra contained only noise. See the influence of different noise removal techniques on the estimation of high frequency attenuation in our companion paper (Aslan et al., 2020).

Often both $H_{e,cs}$ and $H_{e,ps}$ are approximated by $H$ with a value for $\tau$ that describes the high-frequency attenuation of the

EC setup according to Eq. (5). Then the measured flux is corrected for high-frequency attenuation by multiplying it with CF (see Eq. (1)). There are different ways to estimate $Co^{w,\chi}$ for Eq. (1). Here we limited the analysis only to periods when the absolute sensible heat fluxes were higher than 15 W m$^{-2}$ and estimated $Co^{w,\chi}$ in Eq. (1) with the measured $Co^{w,T}$ following Fratini et al. (2012).

We used four methods to estimate EC system response times (and additionally $t_{lpf}$ in Method 4) and then to calculate

$CF$ (see Table 1). Method 1 follows the ICOS EC data processing protocol (Sabbatini et al., 2018) and is implemented e.g. in EddyPro after Hunt et al. (2016) (see also https://www.licor.com/env/support/EddyPro/topics/whats-new.html). Method 2 follows Aubinet et al. (2000) and is implemented in EddyUH (Mammarella et al., 2016). Method 3 follows Fratini et al. (2012), and Method 4 is the only one that explicitly accounts for the temporal lag caused by the high-frequency filtering action of the system and is introduced in this study. Throughout the study, cross-covariance maximisation was used to account for the

lag between scalar and vertical wind speed as typically done in the global flux measurement network.





### 3.2.2 Low-pass filtering of temperature data

In order to evaluate the performance of the different spectral correction methods presented in Sect. 3.2.1, high frequency $T$ time series were attenuated with different values of $\tau$ in order to mimic attenuated scalar time series. Similar to the procedure used by Aslan et al. (2020), $T$ time series were converted to frequency domain via a Fourier transform, multiplied by Eq. (3)

and converted back to time domain with the inverse Fourier transform. Three different values for $\tau$ were used: 0.1 s, 0.3 s and 0.7 s. The correct value for $CF$ was obtained from the ratio between $\overline{w'T'}$ and $\overline{w'\hat{T}'}$, where $\hat{T}$ was the time series attenuated with the methodology described above. Note that here $\overline{w'\hat{T}'}$ was calculated using cross-covariance maximisation in order to mimic regular EC gas flux processing, meaning that the lag caused by low-pass filtering was taken into account. This correct value for $CF$ was then compared against $CF$ estimated with the four methods described in Sect. 3.2.1 in order to evaluate the

accuracy of the methods.

## 4 Results and discussion

### 4.1 Time lag dependence on attenuation and turbulence time scales

In Sect. 2.2, we explained why the following approximate relationship must hold up to some bound at the high frequency end: $HH_p \approx \sqrt{H}$ from which follows: $\cos\left(-2\pi f t_{lpf}\right) - \omega\tau\sin\left(-2\pi f t_{lpf}\right) \approx \sqrt{1+(2\pi f\tau)^2}$. Assuming that the time lag is

proportional to the time constant, i.e. $t_{lpf} = C\tau$, this equation can be solved numerically for the constant $C$ as a function of frequency $f$. Fig. 1 (upper plot) illustrates that such solution is dependent on frequency and varies over a relatively large interval up to the frequency $\frac{2}{3}\frac{\pi}{\tau}$ (the relationship holds well up to such frequency for any $\tau$). In accordance with the approximations with the Taylor expansion (Sect. 2.2), at low frequencies the constant $C$ equals one. The lower plot of Fig. 1 presents the left and right side terms of the approximate equality using the constant coefficient $C = 0.63$. In spite of the large variation of the

exact coefficient with frequency, the selected constant value produces very good correspondence for the whole frequency range. A value of $C = 0.63$ was obtained by estimating the average time lag in a least square sense over equally spaced frequencies in logarithmic scale over the interval from $\tau = 0.01\frac{\pi}{\tau}$ to $\frac{2}{3}\frac{\pi}{\tau}$ for a range of time constants from 0.05 to 1 s. We observed that the optimum coefficient was independent of $\tau$ and equal to 0.63 (with 2-digit accuracy). Note that these analyses were independent of the actual turbulent signal as they were obtained using the transfer functions only ($H$ and $H_p$).

In addition to the analysis above, the integral in Sect. 2.2, i.e. Eq. (9), was solved numerically with various combinations of time lag, $\tau$ and $(z-d)/U$ in order to evaluate which time lag value resulted in maximum value for the integral for a given $\tau$ and $(z-d)/U$ combination. This identified the dependence of the low-pass filtering induced lag ($t_{lpf}$) on $\tau$ and the turbulence time scale ($(z-d)/U$) and adds to the analysis above by incorporating the influence of turbulent signal on the dependence. For this numerical experiment, $\tau$ varied between 0.05 s and 1 s, $(z-d)/U$ between 1 s and 10 s and $Co$ was calculated based on a

model (similar to the one in Kristensen et al. (1997) but fitted to Siikaneva observations). Based on this analysis, $t_{lpf}$ depended approximately linearly on $\tau$ (Fig. 2) in accordance with the analysis above, yet the dependence had an additional weak $\frac{\tau U}{z-d}$



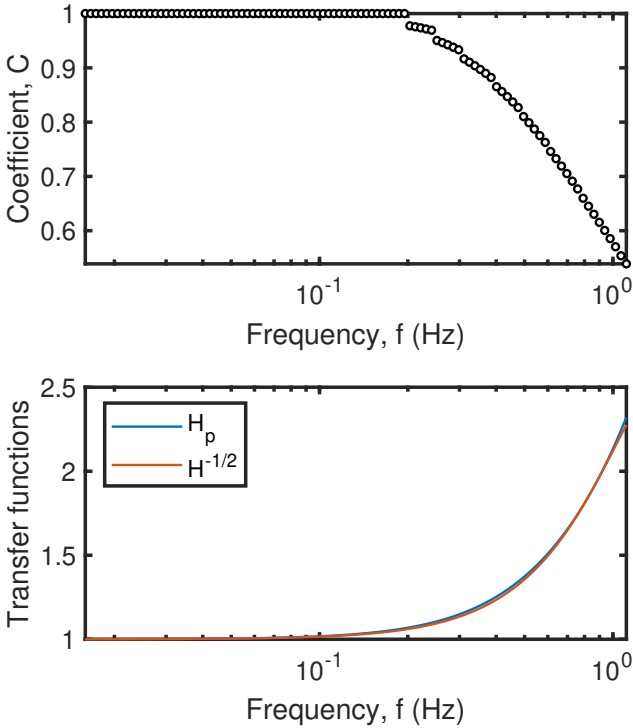

**Figure 1.** Top: the value for the proportionality constant $C$ ($t_{lpf} = C\tau$) obtained when approximating $H_p \approx 1/\sqrt{H}$ at different frequency ranges. Bottom: $H_p$ and $1/\sqrt{H}$ calculated using the proportionality $t_{lpf} = 0.63\tau$. See more details in Sect. 4.1.

term:

$$t_{lpf} = 0.73\tau - 0.15\,s\frac{\tau U}{z - d} + 0.02\,s(R^2 = 0.998). \tag{14}$$

However, this numerical experiment was repeated also for very high attenuation levels and the found dependence above failed to fully recover $t_{lpf}$ values at high attenuation and this bias increased with $\tau$ and inverse of turbulence time scale (i.e. $U/(z-d)$).

5   For example at $\tau = 2$ s and $(z-d)/U = 1$ s the equation above gave $t_{lpf} = 1.2$ s, whereas based on the numerical integration value 1.0 s was obtained, meaning that the dependence above was not able to accurately describe the low-pass induced time lag at these high attenuation levels. This is likely due to the fact that also the shape of the cospectrum $Co$ in Eq. (9) has an effect on $t_{lpf}$ dependence on $\tau$. At very high attenuation levels also the peak of the cospectrum is attenuated and this results in a different $t_{lpf}$ dependence on $\tau$ than the equation above which describes the dependence when attenuation takes place at high

10   frequencies (i.e. inertial subrange).

The discrepancy between the two dependencies between $t_{lpf}$ and $\tau$ obtained above was likely due to the fact that the latter took into account also the variability of the turbulent flux with frequency, whereas the former was based purely on transfer functions. Nevertheless, these results indicate that, for a given site, $t_{lpf}$ can be approximated to be constant at a specific

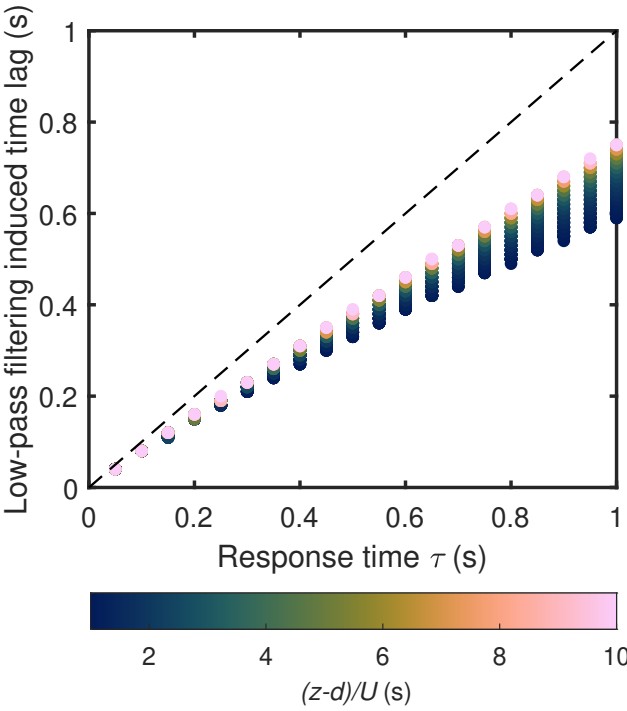

**Figure 2.** Low-pass filtering induced time lag ($t_{lpf}$) plotted against response time ($\tau$). $t_{lpf}$ values were estimated by numerically integrating the integral in Sect. 2.2 with various time lag values and evaluating which value resulted in maximum value for various $\tau$ and $(z-d)/U$ combinations. The dashed line shows 1:1 correspondence. See more details in Sect. 4.1.

attenuation ($\tau$) level and hence it can be used as a fitting parameter when evaluating the high frequency response of the EC system (see Sect. 3.2.1, Method 4).

## 4.2 Assessment of the flux loss correction methods using an attenuated T time series

The four methods to estimate the CF (Table 1) were evaluated using an artificially attenuated turbulent $T$ time series comparing
5   different values for $\tau$. Figure 3 shows example cross-covariance functions from the Siikaneva site with three different values of $\tau$ applied to the same 2-hour time series. The peak of the $w$-$\hat{T}$ cross-covariance shifted to longer positive times (i.e. $\hat{T}$ lags $w$) and the cross-covariance maximum decreased as $\tau$ increased. This is related to the phase shift and flux attenuation caused by the low-pass filter, respectively. The low-pass filtering was also evident in the corresponding power spectra and cospectra. The resulting empirical transfer functions in Fig. 4 show that power spectra were more attenuated than the cospectra, when
10  cospectra were calculated from data where the time lag derived from cross-covariance maximisation was used to shift the time series (as is typically done when processing EC data). Moreover, the shape of the empirical transfer function derived from the cospectra differed from Eq. (5). This difference in shape was related to $H_p$ and the phase shift caused by the low-pass filter. Note that the phase shift resulted in negative values for $HH_p$ and $H_{e,cs}$ at high frequencies, meaning that the related





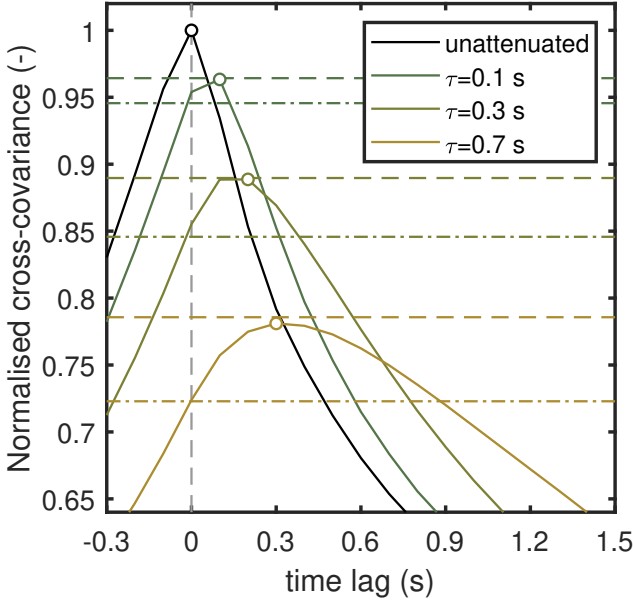

**Figure 3.** Cross-covariance between $w$ and $\hat{T}$ calculated from data measured between 9:30 and 11:30 16.6.2013 at Siikaneva site. $T$ data were attenuated with Eq. (3) and three different values for $\tau$ in order to demonstrate the effect of filtering on cross-covariance. All the cross-covariance values were normalised with maximum of unattenuated cross-covariance (i.e. unattenuated covariance). Dots highlight the maxima. Horizontal dashed lines show the attenuation factor (inverse of correction factor, $1/CF$) values calculated with Method 4 and dash-dot lines show the attenuation factors calculated with Method 1. Note that the dash-dot lines match the cross-covariance values at zero lag, whereas dashed lines agree with the cross-covariance maxima.

attenuated cospectrum had both positive and negative values. This implies that attenuated cospectra should not be presented on log-log scales (negative values cannot be presented on log-scale). Note also that $HH_p$ is not exactly equal to $\sqrt{H}$ at all frequencies. This relates to the fact that $Q_m$ cannot be nullified with a constant time shift since low-pass filtering induces a frequency-dependent time shift ($\frac{\arctan(-\omega\tau)}{\omega}$, see Massman (2000)). If $Q_m$ could be nullified, then $Co_m = A_m$ (see Eq. (10))

5  and the attenuation of cospectra could be described accurately with $\sqrt{H}$.

When the correction factor was calculated with Method 4, which takes into account the low-pass filtering induced phase shift, the attenuation factors ($1/CF$) agree with the normalised cross-covariance maxima (see Fig. 3). This indicates that the attenuation of cross-covariance maxima was accurately estimated with this method. However, when $H_p$ was neglected and the attenuation factors were calculated with $H$ only, then they agreed with cross-covariance values at zero lag as per predicted

10  by theory (Sect. 2). Note that the time lag discussed here only represents the low-pass filter effect, not other effects that also alter the time lag. This is only possible because we work with degraded temperature time series, where phase shifts through low-pass filtering are the only cause for time delays.

The value for $\tau$ used in attenuating the T time series was further varied between 0.05 s and 1 s and used for the same 2-hour time series. The bias in CF calculated with Method 1 increased with $\tau$ (Fig. 5a). The other methods did not produce





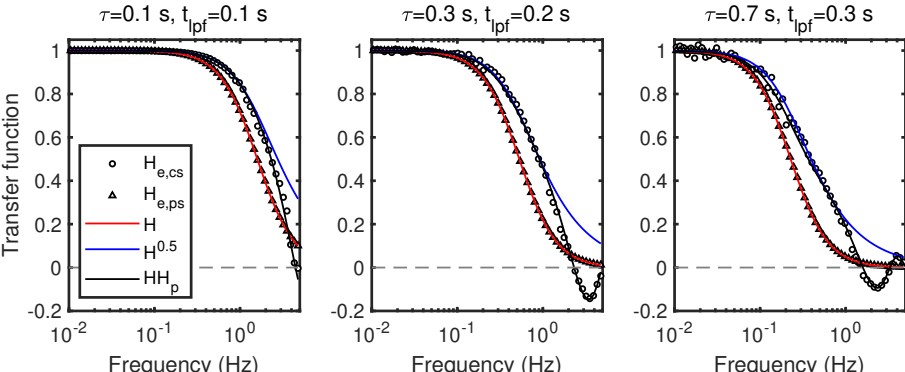

**Figure 4.** Transfer functions for the same data shown in Fig. 3. Note that here $H$, $\sqrt{H}$ and $HH_p$ were not fitted but calculated using the known values for $\tau$ and $t_{lpf}$ resulting from the low-pass filtering of the $T$ time series (Sect. 3.2.2).

as significantly biased estimates for CF when using this example dataset, although Methods 2 and 3 somewhat underestimated $CF$ and this underestimation increased with $\tau$. These findings are in agreement with Fig. 4: $\sqrt{H}$ was above and $H$ below the empirical transfer function calculated from the cospectrum ($H_{e,cs}$), meaning that they underestimated and overestimated the attenuation, respectively. These findings are also in accordance with Aslan et al. (2020) results with noisy measurements. Note

that Method 2 estimated close to correct value for $CF$ (Fig. 5a) despite biased estimates for the response time (Fig. 5b) due to the compensation of two errors (biased $\tau$ and incorrect shape for the cospectral transfer function). Method 4 was able to reproduce the correct value for $\tau$ from $H_{e,cs}$ (Fig. 5) indicating that the difference between $H_{e,cs}$ and $H_{e,ps}$ was indeed related to $H_p$ and the low-pass filtering related phase shift. The time lags estimated by Method 4 (i.e. by fitting $HH_p$ to $H_{e,cs}$) agreed with the ones estimated by maximising the crosscovariance (Fig. 5c) also corroborating this conclusion. It should be noted that

the step changes in the data that are very visible in Fig. 5c but to some extent also in Figs. 5a and b are caused by the finite resolution of the underlying data of $w$ and $T$ of 0.1 s.

In order to evaluate the four methods to estimate CF further, turbulent $T$ data were low-pass filtered with three values of $\tau$ (0.1 s, 0.3 s and 0.7 s) over several summer months (May-August) at two contrasting EC sites, Hyytiälä and Siikaneva. Then the related flux attenuation was corrected with the four methods and the results were compared against a reference flux calculated

from unattenuated $T$ data. Table 2 summarises the relative differences between the four correction methods and the reference. The performance of all four methods decreases with increasing $\tau$ at both sites, and Method 1 showed the worst performance, which is in line with the example shown in Fig. 5 and follows the predictions from theory (Sect. 2). Of the analysed cases, the biggest bias (+8.6 %) was found when applying Method 1 at Siikaneva with $\tau = 0.7\,s$. In general, the performance of the methods was worse at Siikaneva than at Hyytiälä. This was likely related to the lower measurement height at Siikaneva

($z - d$=2.7 m) than at Hyytiälä ($z - d$=13 m) and thus the comparably larger contribution of high frequencies to the covariance. At Siikaneva, smaller eddies dominated the turbulent transport and hence the effect of inaccuracies in describing the high frequency attenuation of the signal was amplified compared with Hyytiälä.





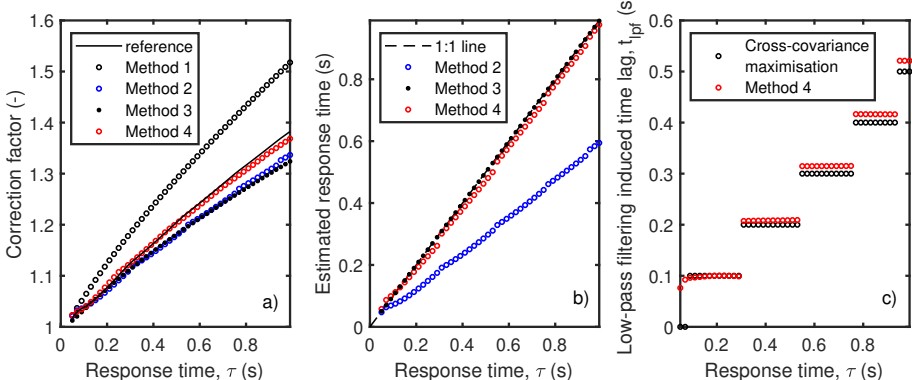

**Figure 5.** Comparison of estimated correction factors and response times to the values used to attenuate the $T$ time series. The reference values for the correction factors were calculated as the ratio between $\overline{w'T'}$ and $\overline{w'\hat{T}'}$, where $\hat{T}$ is the temperature time series attenuated according to Sect. 3.2.2. Note that $\overline{w'\hat{T}'}$ was calculated using cross-covariance maximisation. The time lags obtained from cross-covariance maximisation and from fitting $HH_p$ to $H_{e,cs}$ are shown in panel (c). The same data were used as in Figs. 3 and 4.

**Table 2.** Mean relative difference between the fluxes based on the different correction methods applied to sensible heat fluxes attenuated with different values for $\tau$ ((cor-ref)/ref*100 %) and the reference (unattenuated sensible heat flux). All values are in %.

| Site | Prescribed $\tau$ | Method 1 | Method 2 | Method 3 | Method 4 |
|------|------------|----------|----------|----------|----------|
| Hyytiälä | 0.1 s | +0.6 | +0.2 | +0.1 | +0.3 |
| Hyytiälä | 0.3 s | +1.7 | +0.6 | +0.5 | +0.7 |
| Hyytiälä | 0.7 s | +3.8 | +1.0 | +1.0 | +1.5 |
| Siikaneva | 0.1 s | +2.4 | -0.1 | +0.2 | -0.05 |
| Siikaneva | 0.3 s | +5.2 | -0.5 | -0.4 | +0.3 |
| Siikaneva | 0.7 s | +8.6 | -1.4 | -1.5 | +0.2 |

The importance of turbulence scale on the performance of the four methods was evaluated by stratifying the data according to stability and plotting them against $\tau/\text{ITS}$ (Fig. 6) where ITS is the integral time scale of $w'T'$, calculated from the auto-covariance of $w'T'$ time series by integrating the normalised autocovariance function until its first zero crossing (e.g. Kaimal and Finnigan, 1994). This ratio describes the ratio of attenuation and turbulent transport scales. It is also worth noting that this ratio can alternatively be approximated by the ratio of the cut-off frequency ($f_{co} = 1/(2\pi\tau)$) to the cospectral peak frequency ($f_m = n_m U/(z-d)$, where $n_m$ is the normalised cospectral peak frequency) (Rannik et al., 2016). High values for $\tau/\text{ITS}$ describe periods when the attenuation time scale is large relative to the time scale of turbulent scalar transport, whereas low values correspond to cases when the attenuation time scale is small relative to this turbulence scale. When CF was estimated with Methods 2 or 3, the relative bias in CF was typically within $\pm 3\,\%$ and increased linearly with $\tau/\text{ITS}$ at Siikaneva. For





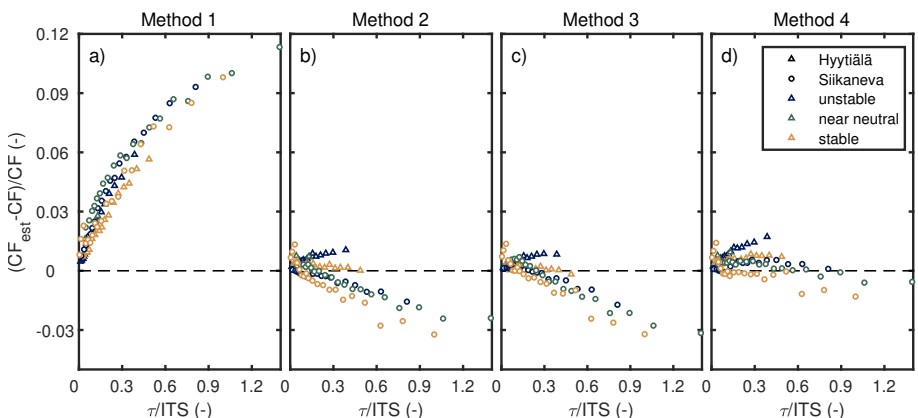

**Figure 6.** Relative bias in the estimated correction factor ($CF_{est}$) against $\tau/ITS$ where ITS is the integral time scale of $w'T'$. Data were divided into unstable ($\zeta < -0.1$), stable ($\zeta > 0.1$) and near-neutral ($|\zeta| < 0.1$) cases before bin-averaging and plotting. Reference values for the correction factor (CF in the figure) were calculated from the ratio between unattenuated and attenuated covariances.

Method 4 the bias was within $\pm 2\%$ at both sites and independent of the $\tau/ITS$ ratio. The remaining small bias obtained with Method 4 can be speculated to be related to the quadrature spectra, which was assumed to be zero, albeit strictly speaking it is not (Aslan et al., 2020). For Method 1 the relative bias in CF scaled with $\tau/ITS$ and after normalisation with ITS the data from both field sites collapsed into one common relationship (Fig. 6a). Horst (1997) derived a simple formula to estimate CF for

different situations as a function of $(\tau/ITS)^{\alpha}$, where $\alpha = 7/8$ in neutral and unstable cases and $\alpha = 1$ in stable cases. Similar dependencies could be derived for the bias in CF derived with Method 1.

    Since the relative bias in CF when estimated with Method 1 scales with $\tau/ITS$, this dependence can be used to assess how much scalar fluxes are biased with different EC setups when the Method 1 is utilised in the EC data processing. For the following example calculations we approximate a linear dependence between ITS and $(z-d)/U$ and used the fit to estimate ITS

for different $z-d$ and $U$ combinations. For a short tower ($z-d$=1.5 m) with moderate wind speed ($U = 2$ m/s) and attenuation ($\tau = 0.2$ s), fluxes are biased by 4% based on the dependency shown in Fig. 6a. For the same tower, higher attenuation ($\tau = 0.8$ s) results in larger bias (9 %) if the Method 1 is utilised. For a tall tower the biases are not as significant since larger eddies dominate the transport and hence $\tau/ITS$ is smaller for a given $\tau$ than in case of a shorter tower. For a tall tower ($z-d$=20 m), with moderate wind speed ($U = 3$ m/s) and attenuation ($\tau = 0.2$ s) the bias is 1-2% and with higher attenuation ($\tau = 0.8$ s) the

bias is 3-4%. These example calculations demonstrate the magnitude of the flux bias resulting from biased spectral corrections at contrasting EC sites.

### 4.3   CO$_2$ and H$_2$O fluxes corrected with the four methods

The applicability of the results acquired with attenuated $T$ was analysed by processing $CO_2$ and $H_2O$ fluxes from summer months at the two sites (Sect. 3.1). The gas fluxes were corrected with the four spectral correction methods in order to assess

the systematic biases stemming from biased spectral corrections. For $CO_2$ the fitting of $H$ to $H_{e,ps}$, of $H$ to $H_{e,cs}$ and of





$HH_p$ to $H_{e,cs}$ estimated response times of 0.05 s, 0.10 s and 0.13 s at Hyytiälä and of 0.13 s, 0.12 s and 0.18 s at Siikaneva, respectively. The low-pass induced time lag ($t_{lpf}$) estimated as a fitting parameter when fitting $HH_p$ to $H_{e,cs}$ yielded close to 0.1 s at both sites for $CO_2$. For reference, the lag times estimated via cross-covariance maximisation which are combinations of signal travel times in the sampling line ($t_{phys}$) and $t_{lpf}$ yielded on average 1.4 s and 0.2 s at Siikaneva and Hyytiälä,

respectively.

Based on theoretical considerations (Sect. 2) and results obtained with attenuated $T$ time series, $H$ fit to $H_{e,cs}$ (Method 2) was projected to estimate smaller response time than the other two methods that were projected to agree (see e.g. Fig. 5b). Instead, here for $CO_2$ the $HH_p$ fit to $H_{e,cs}$ gave longer response times than the other two methods (Fig. 7). For signals with small attenuation both $\tau$ and $t_{lpf}$ are likely difficult to estimate accurately by fitting $HH_p$ to $H_{e,cs}$. The low-pass filtering

induced lag ($t_{lpf}$) can attain values only with the temporal resolution of the underlying data itself (here 0.1 s) and hence for time series exhibiting small attenuation $t_{lpf}$ may well be zero. This would cause $H_{e,ps}$ and $H_{e,cs}$ to be similar and hence they would result in similar estimates for the response time, as observed here for $CO_2$. This granular estimation of $t_{lpf}$ might explain why the response times for $CO_2$ did not follow the expectations. It should be also noted that the methods to estimate the response time have a limited accuracy of estimation stemming from the algorithms used in the estimation, instrument limitations and

meteorological conditions under which the observations are made (Aslan et al., 2020). Furthermore, the empirical transfer function derived from cospectra ($H_{e,cs}$) contains the attenuation related also to sensor separation (Horst and Lenschow, 2009), whereas the empirical transfer function estimated from power spectra ($H_{e,ps}$) is related only to gas analyser induced low-pass filtering.

For $H_2O$, $\tau$ depends on RH (Mammarella et al., 2009; Ibrom et al., 2007a; Runkle et al., 2012) and hence the analysis

was done in RH bins. Figure 8 shows an example of empirically derived transfer functions for $H_2O$ at Siikaneva in moderate relative humidity conditions. $H_{e,cs}$ showed less high frequency attenuation than $H_{e,ps}$ due to cross-covariance maximisation as predicted by the theory (Sect. 2). $HH_p$ fit to $H_{e,cs}$ resulted in similar value for $\tau$ as the one obtained by fitting $H$ to $H_{e,ps}$. The other fitting parameter in $HH_p$ fit ($t_{lpf}$) was 0.14 s indicating the low-pass filtering of $H_2O$ time series caused an additional lag to the $H_2O$ time series. Note that $H$ does not perfectly describe the shape of $H_{e,ps}$. This is due to adsorption and desorption of

$H_2O$ to sampling tube walls, a process which cannot be described accurately with Eq. (3) (Nordbo et al., 2013). Hence different transfer functions have been proposed for $H_2O$ (De Ligne et al., 2010; Nordbo et al., 2014).

Figure 9 shows results obtained for $H_2O$ in different RH bins at the two sites. The $H$ fit to $H_{e,cs}$ gave smaller values for $\tau$ in agreement with the results obtained with attenuated $T$ (Sect. 4.2) and expectations based on the theory (Sect. 2). The $HH_p$ fit to $H_{e,cs}$ was able to reproduce the $H_2O$ time lag dependency on RH, yet it estimated systematically smaller values for the

time lag difference between $H_2O$ and $CO_2$ than obtained with cross-covariance maximisation. The time lag difference between $H_2O$ and $CO_2$ can be assumed to be related to $t_{lpf}$ of $H_2O$ since $H_2O$ is more attenuated than $CO_2$. The difference between $H_2O$ and $CO_2$ time lags obtained with cross-covariance maximisation showed linear dependence on estimated response times in relative humidity bins with a similar dependency at both sites (y=0.47x-0.05 s for Siikaneva and y=0.44x-0.07 s for Hyytiälä, where y is the time lag difference and x response time. Fits were based on orthogonal linear regression). Difference between

$H_2O$ and $CO_2$ time lags resulted from stronger low-pass filtering of $H_2O$ than $CO_2$ signal and this analysis assumes that the





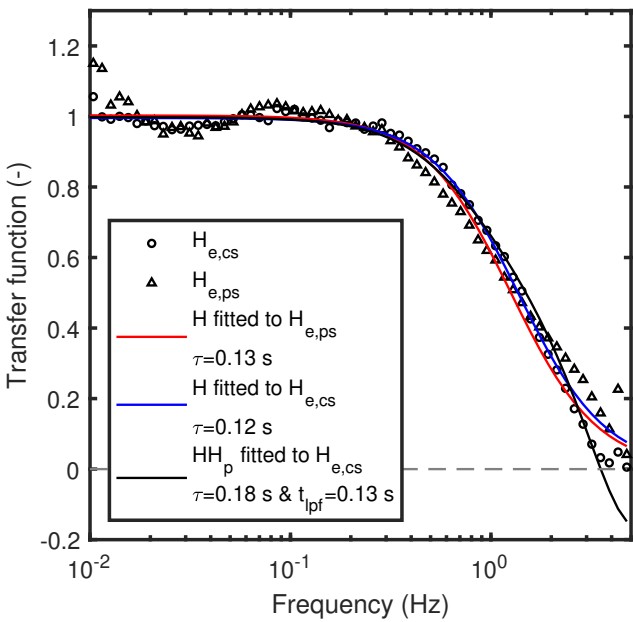

**Figure 7.** Empirical transfer functions estimated for $CO_2$ from ensemble averaged cospectra ($H_{e,cs}$, Eq. (12)) and power spectra ($H_{e,ps}$, Eq. (13)) and three fits to the empirical transfer functions. Data from Siikaneva were used.

mere travel times through the sampling lines (i.e. $t_{phys}$), which do not depend on the low-pass filtering effects, were the same for $H_2O$ and $CO_2$. These results are in line with Sect. 4.1 where it was shown that the low-pass filtering induced time lag was primarily determined by a linear dependence on $\tau$ with a secondary dependence on turbulence time scale. However, the slopes obtained above were different from the ones derived in Sect. 4.1.

5      The $CF$ values estimated with the four methods agreed at Hyytiälä for low RH periods, but they diverged at high RH periods. For instance when RH>80 % the four methods gave on average 1.39, 1.29, 1.23 and 1.37 at the Hyytiälä site, respectively. These results are in line with the ones obtained with attenuated T time series (Sect. 4.2): when $\tau$ was large, method 1) overestimated and method 3) underestimated $CF$ similarly as here for $H_2O$ at high RH (and hence large $\tau$). In contrast, for Siikaneva Method 1 gave systematically higher value for the correction factor regardless of RH, whereas the other three methods more similar CF

10    RH dependency. This difference to Hyytiälä was likely due to lower $\tau$ values also at high RH (Fig. 9). Note also that the EC measurement setup (measurement height, gas sampling system) was not identical at the two sites.

On average, the differences between the four different methods to calculate $CF$ were small, typically within $\pm 3\%$ at both sites for both gases ($CO_2$ and $H_2O$) (Table 3). The biggest mean relative difference (4.1%) was found for $H_2O$ measurements at Siikaneva site for Method 1. This was likely due to the combination of low measurement height and high attenuation which

15    resulted in biased CF values, in accordance with the findings obtained with the attenuated $T$ time series (Fig. 6a). Interestingly, for $CO_2$ the biggest difference at both sites was obtained with Method 3, yielding smaller fluxes than the reference (Method 4). The difference was bigger than the one obtained with Method 1 which contradicts the findings presented above with attenuated

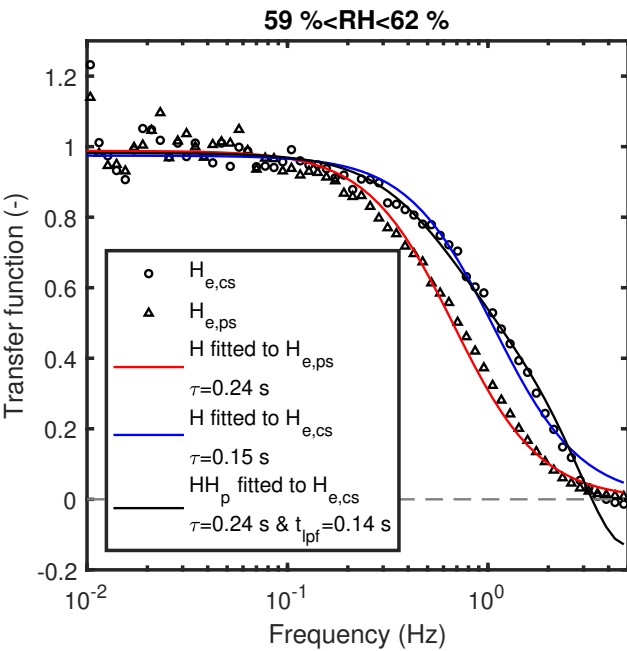

**Figure 8.** Similar figure as Fig. 7 but for $H_2O$ in moderate relative humidity conditions (59%<RH<62%). Note that $\tau$ estimated from $H_{e,ps}$ agrees with $\tau$ estimated from $H_{e,cs}$ with $HH_p$. Data from Siikaneva were used.

**Table 3.** Mean relative difference between the different correction methods applied to $CO_2$ and $H_2O$ fluxes ((cor-ref)/ref*100 %). Fluxes corrected with Method 4 was used as a reference for the other three methods. All values are in %.

| Site | Gas flux | Method 1 | | | Method 2 | | | Method 3 | | |
|------|----------|-----|-------|-----|------|-------|-----|------|-------|-----|
| | | All | Night | Day | All | Night | Day | All | Night | Day |
| Hyytiälä | $CO_2$ | -0.5 | -0.7 | -0.5 | +0.07 | +0.1 | +0.08 | -0.7 | -1.0 | -0.7 |
| Hyytiälä | $H_2O$ | +0.7 | +1.6 | +0.7 | -1.0 | -1.4 | -1.0 | -2.3 | -4.0 | -2.3 |
| Siikaneva | $CO_2$ | +0.4 | +0.3 | +0.4 | -0.3 | -0.2 | -0.3 | -2.4 | -1.9 | -2.4 |
| Siikaneva | $H_2O$ | +4.1 | +4.7 | +4.1 | -0.6 | -0.8 | -0.6 | -1.0 | -1.0 | -1.0 |

$T$ (Sect. 4.2) and expectations based on theory (Sect. 2). Infrared gas analysers for $CO_2$ and $H_2O$ can often be mounted close to the inlet and in both setups considered here inlet lines are fairly short. The measurement of other gases, incl. $CH_4$ and $N_2O$, usually require larger equipment which often has to operated on the ground and/or some distance away from the mast, requiring longer inlet lines with slow time-response. In these situations the variability between methods is expected to be larger.





**Figure 9.** Response times (top row), low-pass filtering induced time lags (middle row) and correction factors (bottom row) as a function of relative humidity. Results for Hyytiälä pine forest are on the left and for Siikaneva open peatland on the right. The lines describe the fits used to obtain $\tau$ and $t_{lpf}$ values for the $CF$ calculation (see Sect. 3.2.1).





### 4.4 Correct form for cospectral transfer function

There has been a long-standing debate about what is the correct form for cospectral transfer function when the scalar measurements are done with first-order linear sensor and vertical wind speed is not attenuated. The seminal paper on EC frequency-response corrections by Moore (1986) described the flux attenuation related to the scalar sensor with $\sqrt{H}$ (see Eq. (5)), which

was later deemed erroneous, e.g. by Eugster and Senn (1995); Horst (1997, 2000) and others. Partly based on similar derivations as shown here in Sect. 2, the correct form for the cospectral transfer function was argued by Horst (1997, 2000) and Massman (2000) to be $H$, instead of $\sqrt{H}$. Lately, both forms ($H$ and $\sqrt{H}$) have been utilised in the literature (e.g. Fratini et al., 2012; Foken et al., 2012b; Mammarella et al., 2016; Hunt et al., 2016; Wintjen et al., 2020) without clear consensus on which one is correct. Part of this confusion is likely related to the fact that some of the past studies have been using incorrect

terminology by using the term cospectral transfer function (Eq. (5)), even though they have derived/utilised transfer function for the amplitude spectrum (Eq. (10)).

Briefly summarising the findings in this study, it was shown in Sect. 2 in accordance with e.g. Horst (2000) that in the case the travel time in the gas sampling system could be accurately estimated and accounted for, the attenuation of cospectrum (and hence flux) could be described with $H$. However, cross-covariance maximisation inadvertently accounts also for the scalar

low-pass filtering related time shift ($t_{lpf}$) and consequently induces a phase shift between the lag corrected attenuated scalar and vertical wind time series. Hence in the case that cross-covariance maximisation is used to align the two time series, the correct form for cospectral transfer function is $HH_p$, where the part related to the phase shift ($H_p$) is calculated with $t_{lpf}$. It was shown above that $HH_p$ can be approximated with $\sqrt{H}$. Therefore the debate about which is the correct form for cospectral transfer function ($H$ or $\sqrt{H}$) relates to the low-pass filtering induced phase shift as discussed already by Eugster and Senn

(1995) and Horst (2000), albeit with a different conclusion.

### 5 Summary and conclusions

The influence of low-pass filtering induced phase shift on estimation of high frequency response of an EC setup was analysed. The analysis assumed that the EC setup consisted of a fast-response anemometer and a linear, first-order-response scalar sensor. Spectral corrections aiming at correcting the EC fluxes for high frequency attenuation were estimated with four methods: three

widely used methods and one newly proposed method which specifically accounts for the interaction between the low-pass filtering induced phase shift and high frequency attenuation. Based on theoretical considerations and experimental results we come to the following conclusions:

- Cross-covariance maximisation overestimates the signal travel time in the EC sampling line since it inadvertently accounts also for the lag caused by low-pass filtering of scalar time series caused by non-ideal measurement system. The
bias in the estimated time lag depends linearly on low-pass filter response time ($\tau$) with a small additional dependence on turbulence time scale.





- Both power spectra and cospectra are attenuated with the same transfer function ($H$, as noted also by Horst (2000)) in the case that the travel time of the scalar signal in the sampling line can be accurately estimated. However, if cross-covariance maximisation is used, then attenuation of cospectra follows $HH_p$ where $H_p$ accounts for the bias in the estimated travel time caused by cross-covariance maximisation and it was shown that $HH_p$ can be approximated with $\sqrt{H}$. Both $H$ and $\sqrt{H}$ have been previously used in the literature to describe the attenuation of cospectra (Moore, 1986; Eugster and Senn, 1995; Horst, 1997; Moncrieff et al., 1997; Horst, 2000; Massman, 2000; Massman and Lee, 2002; Ibrom et al., 2007a; Mammarella et al., 2009; Fratini et al., 2012; Wintjen et al., 2020), yet clear consensus on which one is the correct form has not been established. Here we show that $\sqrt{H}$ approximates the correct form (i.e. $HH_p$), whereas $H$ is incorrect if cross-covariance maximisation is used.

- In order to estimate and correct for the flux attenuation correctly, it is vital to accurately describe the attenuation of the cospectra in the correction procedure. Hence, while fitting $H$ to cospectra (Method 2) biased the first-order response time due to the use of the incorrect cospectral transfer function, it resulted only in a small bias for the flux correction factor, since the method describes the attenuation of cospectra accurately once the cross-covariance maximisation has been applied. By contrast, fitting of $H$ to power-spectra (Method 1) correctly estimates the response time, but it nevertheless yields biased flux corrections since it utilised a transfer function that is incorrect when estimating the flux using cross-covariance maximisation. The bias in EC fluxes can be 10% or above if method 1) is used for short tower measurements with moderate attenuation.

- All flux calculation algorithms which rely on cross-covariance maximisation and at the same time use $H$ to describe the attenuation of the cospectra can be projected to be biased in the same way as Method 1 used here if the response time (i.e. $\tau$) is accurately estimated (e.g. from power spectra). Hence, for instance the analytical formulas of Horst (1997) and Massman (2000) will result in biased fluxes when an accurate estimate of scalar sensor response time is used in these algorithms.

- The theoretical framework proposed in this paper was able to describe the changes in $H_2O$ time lag as $H_2O$ response time increased with relative humidity. This suggests that the findings derived here with attenuated temperature time series are applicable also in real world situations for EC gas flux measurements. Similarly, Wintjen et al. (2020) showed that the transfer function related to the phase shift ($H_p$) needed to be taken into account when processing reactive nitrogen fluxes. However, results for $CO_2$ deviated from the expectations made based on the theory. This might be due to $CO_2$ response time being small and thus close to the data temporal resolution (0.1 s) which resulted in difficulties in observing the low-pass filtering induced time lag. Alternatively, low-pass filtering of inert gases (such as $CO_2$) in the gas sampling line may deviate from what was described here in such a way that the low-pass filter acting on $CO_2$ may not induce a phase shift. This might be the case if the system's high frequency attenuation is dominated e.g. by volume averaging in the gas sampling cell (Massman, 2004). This should be investigated by using $CO_2$ flux data from an EC setup with pronounced attenuation of $CO_2$ fluctuations in the sampling line. Also, the recent laboratory studies on the subject (Metzger et al.,





2016; Aubinet et al., 2016) are not helpful in this matter since they did not consider low-pass filtering induced time lags in their analyses.

In summary, it is suggested that the spectral correction methods implemented in EC data processing software are revised so that the influence of low-pass filtering induced phase shift is recognised following the findings presented above.

*Code and data availability.* Data and matlab codes to reproduce Figures 3, 4 and 5 will be uploaded to open data repository Zenodo upon acceptance of this manuscript along with flux time series processed with the four methods (Table 1) at the two sites.

**Appendix A: Derivation of Eq. (4)**

Here we derive Eq. (4) starting from the cross-spectrum (Eq. (2)):

$$Cr_m = [Z_w][h_\chi Z_\chi]^* e^{-j\phi_{phys}}, \tag{A1}$$

where $h_\chi$ can be described with Eq. (3), which gives

$$Cr_m = [Z_w]\left[\frac{Z_\chi}{1 - j\omega\tau}\right]^* e^{-j\phi_{phys}} \tag{A2}$$

$$Cr_m = [Z_w]\left[\frac{Z_\chi^*}{1 + j\omega\tau}\right] e^{-j\phi_{phys}} \tag{A3}$$

$$Cr_m = Z_w Z_\chi^* \frac{1}{1 + j\omega\tau} e^{-j\phi_{phys}} \tag{A4}$$

$$Cr_m = \frac{Co + jQ}{1 + j\omega\tau} e^{-j\phi_{phys}} \tag{A5}$$

$$Cr_m = \frac{(Co + jQ)(1 - j\omega\tau)}{1 + \omega^2\tau^2} e^{-j\phi_{phys}} \tag{A6}$$

$$Cr_m = \frac{(Co - j\omega\tau Co + jQ + \omega\tau Q)}{1 + \omega^2\tau^2} e^{-j\phi_{phys}}, \tag{A7}$$

where $Z_w Z_\chi^* = Co + jQ$ was utilised. Next quadrature spectrum ($Q$) was assumed zero (Horst, 1997; Massman, 2000) and hence

$$Cr_m = \frac{Co - j\omega\tau Co}{1 + \omega^2\tau^2} e^{-j\phi_{phys}} \tag{A8}$$

Now we can use Euler formula ($e^{-j\phi_{phys}} = \cos\phi_{phys} - j\sin\phi_{phys}$)

$$Cr_m = \frac{Co - j\omega\tau Co}{1 + \omega^2\tau^2}(\cos\phi_{phys} - j\sin\phi_{phys}) \tag{A9}$$

$$Cr_m = \frac{Co\cos\phi_{phys} - j\omega\tau Co\cos\phi_{phys} - jCo\sin\phi_{phys} - \omega\tau Co\sin\phi_{phys}}{1 + \omega^2\tau^2} \tag{A10}$$

$$Cr_m = \frac{\cos\phi_{phys} - \omega\tau\sin\phi_{phys}}{1 + \omega^2\tau^2}Co - j\frac{\sin\phi_{phys} + \omega\tau\cos\phi_{phys}}{1 + \omega^2\tau^2}Co \tag{A11}$$

which equals the Eq. (4) in Sect. 2.





*Author contributions.* OP devised the original concept for the study and all the authors provided further ideas. OP did the data analysis, with input also from ÜR. OP wrote the first draft of the manuscript, with contributions also from TA, ÜR and IM. All the authors commented the first draft and made improvements.

*Competing interests.* The authors declare that they have no conflict of interest.

5 *Acknowledgements.* This study was supported by ICOS and through the H2020 RINGO project of the European Commission (grant 730944). TA is grateful to the Finnish National Agency For Education and The Vilho, Yrjö and Kalle Väisälä foundation for their kind support for funding. OP is supported by the postdoctoral researcher project (decision 315424) funded by the Academy of Finland, and EN acknowledges support by the Natural Environment Research Council award number NE/R016429/1 as part of the UK-SCAPE programme delivering National Capability.



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
