# Peer review of "The high frequency response correction of eddy covariance fluxes. Part 21: the empirical-an experimental approach and its interdependence with the time-lag estimation"

_Atmospheric Measurement Techniques, 2020_

## Author Comment (AC1)

Manuscript title:The high frequency response correction of eddy covariance fluxes. Part 2: the empirical approach and its interdependence with the time-lag estimation

Authors: Olli Peltola, Toprak Aslan, Andreas Ibrom, Eiko Nemitz, Üllar Rannik, and Ivan Mammarella

MS No.: amt-2020-479

We thank all the three referees for their comments and constructive criticism. Please find below point-by-point responses to the presented critique. Responses to the comments are in red and the corresponding changes to the manuscript are in blue.

**Referee #1: Marc Aubinet**

General comments

This paper discusses different spectral corrections procedures for low pass filtering effects in eddy covariance systems. Despite eddy covariance has become the most common approach to determine, among others, CO2, water vapour or greenhouse gas budgets of ecosystems, the method still suffers from uncertainties due to random and, more worryingly to my opinion, to systematic errors. From this point of view each study providing a better understanding of measurements errors and improving correction procedure is welcome.

In that respect, the paper by Peltoli et al is important for at least two reasons: first it points a systematic error actually made by some eddy covariance data treatment softwares (including EddyPro, cf Sabbatini et al., 2018) which definitely requires a correction; secondly it clarifies the question of the cospectra transfer function shape and reconcile theory and observations. It also provides a new method to correct low pass filtering effects but I think that it's robustness and applicability to routine measurements needs still to be proven.

For these reasons, I think that the paper deserves publication. As it is generally well written and structured, I think that only minor revision is required before acceptation.

I would add that this study comforts me in the opinion that, despite the great interest of theoretical studies that help understanding the causes and modalities of low pass filtering by eddy covariance systems, empirical approaches relying as little as possible on theoretical hypotheses remain the most robust ones to apply frequency corrections on routine measurements. In particular, in the present study, the approach deducing a transfer function from cospectra rather than from power spectra (Method 2) remains one of the most robust. The fact that the shape of the transfer function and the time constant are not exact is not very problematic to my opinion, as it does not affect critically the values of the correction factor, which is the target. The Method 4 proposed by the authours could be an interesting alternative, as it also relies on cospectra but uses a different transfer function shape. However, it is more complex as it requires the determination of two parameters (against one for Method 2) and, if the method worked well in the present case where the high frequency

attenuation was artificially introduced, I suspect (and they confirm on P16L9) that disentangling the two time constants could be sometimes difficult, even impossible.

My regret is that the authors do not detail an implementation procedure of Method 4 for routine measurements.

RESPONSE: We thank Marc Aubinet for his positive feedback. Our aim is to provide a code and a small dataset along with the (hopefully) accepted article, so that the readers can test all the four methods introduced in the manuscript. This way the implementation of the Method 4 to eddy covariance (EC) data processing chain should be straightforward.

Specific comments

The paper is the second of a series of two papers on spectral corrections. I was first asked to review the first of them (Aslan et al, also available on AMT discussions) but had to wait the submission of this one to really understand some issues and methodical choices of the Aslan paper. As the present paper appears to me more "standing alone", I suggest to place this one in the first place and the Aslan paper in the second place. This is the order I followed for my reviews.

RESPONSE: Due to this comment we decided to change the order of these two papers.

CHANGES:Changed the manuscript title to "The high frequency response correction of eddy covariance fluxes. Part 1: an experimental approach and its interdependence with the time-lag estimation"

Introduction

The introduction offers a review of the knowledge about spectral corrections. It is clear and highlights the most important points. I have no specific comment about this except two small remarks :

P2L13: I think there's a typo ("contribute" rather than "contributed")

RESPONSE: We assume that this comments refers to P2L30

CHANGES:corrected as suggested

P3L7: Reference to Aubinet is not relevant I think as it refers to the chapter "night flux correction" in my book. I suggest to rather refer to the book itself or to a specific chapter (time lag is evoked in Ch 2 – Munger et al., 2012; Ch 3 – Rebmann et al., 2012 and Ch 4 – Foken et al., 2012).

RESPONSE: Thanks, this erroneous reference was due to reference management software.

CHANGES:Changed the references to the Aubinet et al. (2012) textbook.

Theory

I liked this chapter as it helps me to understand the issue of the cospectral transfer function shape. I must say honestly that I overlooked the debate about the presence or not of a square root in the cospectral transfer function shape (for my defense, I was more concerned in the past by the cospectral – Method 2 - approach than by the spectral approach – Method 3) but, when comparing recently spectral and cospectral approaches on crop sites data, I found a better agreement when applying the square root (Method 3) than not (Method 1), which contradicted the theoretical predictions by Horst (1997), among others. I thus found the explanations given by this paper clever and convincing.

Two remarks, anyway:

P5L27-P6L2: I don't see the interest of presenting the approximation on L29. I tested the equation on L29 and found it fitted quite loosely equation 6. In addition, as I understand, this equation was not used in the paper and equality between tlpf and tau was not assumed further. Maybe could you consider to skip this.

RESPONSE: We would like to keep this approximation here, since it is needed to derive the approximate relationship tlpf ≈ tau. As noted by the referee, this approximation is not very good. However, this is already noted in the manuscript by showing that the proportionality constant C (tlpf=C*tau) depends on frequency (Fig. 1 in the manuscript).

CHANGES:No changes

P5L29: I'm wondering about the equality (and below, the proportionality) between tlpf and tau. Indeed, these two time constants are a priori not physically linked (except when both result from tube attenuation, which is of course an important case) and I'm wondering if you don't loose generality by introducing this dependency. This question is discussed below but, in the end, there is no clear description on how you really implement the transfer function computation: do you fit an equation for H Hp based on equations (5) and (6) ? On equation (5) and those of P5L29? Do you consider tau and tlpf as independent parameters or do you relate them in some way?

RESPONSE: There seems to be some misunderstanding related to the Method 4, in particular how tlpf is derived in the method. The dependencies derived in P5L27-P6L2 and shown in Sect 4.1 are not used in estimating the correction factors with the Method 4. These sections of the manuscript simply describe the dependency between tau and tlpf based on theoretical and empirical findings. In the Method 4, tlpf is estimated simply as a fitting parameter (similarly as tau) when $HH_p$ is fitted to Eq. (12) in the manuscript. See Table 1 in the manuscript. We will try to clarify this in the revised manuscript.

CHANGES:Added the following sentence to P6L2: "Note that the dependency between tlpf and tau derived here and in Sect. 4.1 was not used in the estimation of correction factors (see Sect. 3.2.1), but merely to analyse the factors controlling the dependency." Added the following sentence to P8L19: "In the Method 4, both parameters tau and tlpf, were estimated by fitting $HH_p$ to $H_{e,cs}$." Modified the sentence on P10L13-P11L2: "Nevertheless, these results indicate that, for a given site, tlpf can be approximated to be constant at a specific attenuation (tau) level and hence supporting the estimation of tlpf as a fitting parameter in the Method 4 when fitting $HH_p$ to $H_{e,cs}$ (see Sect. 3.2.1 and Table 1)."

Material and methods:

No specific comments. Clear and well presented. It is important to keep in mind (Sect 3.2.2) that the results presented below are not based on real measurements (I mean the attenuation is artificially provoked and does not reflect real attenuation processes), which is a limitation of the study (but this is well stated in the discussion).

RESPONSE: Thanks. We tried to acknowledge this shortcoming in the text. Note however, that the results obtained with $CO_2$ and $H_2O$ measurements describe the performance of the tested methods with real attenuation processes (that is, $CO_2$ and $H_2O$ signals were not artificially attenuated).

CHANGES:No changes

Results:

P9L13-24: Same remark as above: the proportionality between tau and tlps is clear here as both time constant result from an artificial attenuation but how would this relation look like in the case of measurements with a real attenuation and a real time lag, possibly independent ?

RESPONSE: Please see our response above. Also, for instance $H_2O$ data were not artificially attenuated and hence tlpf and tau in Fig. 9 and discussed in P16L31-P17L4 refer to real conditions without artificial attenuation.

CHANGES:No changes

P9L25-27: I was not sure to understand well: is it an approach that mimics the covariance maximisation procedure? If yes, it could be worth specifying it explicitly.

RESPONSE: Yes, this approach mimics the covariance maximisation procedure. This is mentioned in connection to Eq. (9) but will be restated here.

CHANGES:Added the following sentence to P9L27: "This approach is similar to finding the time lag between signals using cross-covariance maximisation and it identified..."

P10L2: What's the meaning of s in Eq 14 (second, I suppose, but I would not mix symbols and units in a formula).

RESPONSE: Yes, s in Eq. (14) stood for seconds. We modified the equation.

CHANGES:Equation is now described with fit parameters a, b and c and their values are given after the equation.

P10L3-4: This sentence let me hunger. As high attenuation could occur often (especially for gases other than CO2) this question should be clarified. Which attenuation levels do you consider? what is the order of magnitude of the bias? what is the impact of this bias on the next steps (correction factor estimation)?

RESPONSE: We give an example of the bias in the estimated tlpf already now for one tau (z-d)/U combination. As this empirical fit is not used in the Method 4 to estimate correction factors (see responses above), we argue that a detailed description of how well this empirical fit describes tlpf variability at high attenuation levels is not needed.

CHANGES:No changes.

P13 Fig 4: As I understand, the red curve corresponds to Method 1, the blue one to Method 3 and the black one to method 4. Is it correct? A direct reference to the method could facilitate figure reading (and why is method 2 absent from the figure?)

RESPONSE: The referee is correct: red is related to Method 1, blue to Method 3 and black to Method 4. Note however, that the H, $H^{0.5}$ and $HH_p$ shown in the figure were not obtained via fitting to $H_{e,cs}$ and $H_{e,ps}$, but rather they were calculated using the known values for tau and tlpf. This is also the reason why method 2 is absent from the figure.

CHANGES: Added the following sentence to Fig. 4 caption: "Nevertheless, H is related to Method 1, sqrt(H) to Method 3 and $HH_p$ to Method 4 (see Table 1)."

P14L5: isn't it rather by the ratio of cospectral peak frequency to the cut off frequency ?

RESPONSE: Yes, you are right. Thanks for noting this.

CHANGES:Changed sentence on P14L4-P14L6 to "It is also worth noting that this ratio can alternatively be approximated by the ratio of the cospectral peak frequency ($f_m=n_mU/(z-d)$, where $n_m$ is the normalised cospectral peak frequency) to the cut-off frequency ($f_{co}=1/(2\pi tau)$) (Rannik et al.,2016)"

P15Fig6: The legend is not fully clear. I suppose that the symbols refer to the sites and the colours to stability conditions. This should be stated more clearly.

RESPONSE: Yes, you are correct, symbols refer to the sites and colors to the stability conditions.

CHANGES:Added the following sentence to Fig. 6 caption: "Note that the different colors refer to different stability conditions and markers to the different sites."

P15Fig6: I'm intrigued by the curve of Hyytialla in unstable conditions for methods 2, 3 and 4. Why is the bias positive, contrary to other site/conditions? Can you comment on this?

RESPONSE: We do not have a direct answer to this question, however it can be speculated to be related to the actual quadrature spectrum Q which was assumed to be zero throughout the study. Turbulent flow above canopies (Hyytiälä) differs from traditional boundary layer flow (Siikaneva) (e.g. Finnigan, 2000) and these differences may inflict nonzero Q above canopies. Nonzero Q would then inflict biases in the estimated correction factors. Q and phase spectra have been largely overlooked in atmospheric turbulence studies and hence this discussion remains speculative at best. Note that the effect of nonzero Q is already mentioned at P15L1-P15L3. Also Referee #2 had some comments on Fig. 6, see below.

CHANGES:Added the following sentence to P14L9: "The different behaviour of the two sites in Figs. 6b and 6c may be related to differences in spectral characteristics of turbulent transport at these two sites since measurements in Hyytiälä were made above forest (roughness sublayer flow) and in Siikaneva above short vegetation (boundary layer flow)."

I'm also intrigued by the fact that the Method 4 more overestimates the correction factor than Methods 2 and 3 (and thus seems to work less good) at Hyytialla in unstable conditions. Here also I would expect a comment.

RESPONSE: See our response above. Note also the sentence at P15L1-P15L3.

CHANGES:Added the following sentence to P14L9: "The different behaviour of the two sites in Figs. 6b and 6c may be related to differences in spectral characteristics of turbulent transport at these two sites since measurements in Hyytiälä were made above forest (roughness sublayer flow) and in Siikaneva above short vegetation (boundary layer flow)."

P15L2-6: I think that the figure shows clearly that the Method 1 gives different results from the three other methods. To my opinion Methods 2, 3, 4 provide all reasonable estimates of the correction factors while Method 1 biases the correction factors due, as you showed in the theory section, to a misinterpretation of the theory. In this sense, giving a relation to quantify the bias introduced by Method 1 is maybe not very useful. It could appear more clearly that this method is wrong and should be definitely not recommended (which notably implies a modification of the ICOS protocol).

RESPONSE: Yes, we agree. Method 1 should not be recommended and the aim here is not to provide a simple fix for the erroneous correction factor values obtained with the method. Instead we use the tau/ITS dependency merely to illustrate the possible bias in fluxes at different sites (P15L7-P15L16) when processed with the Method 1. We will try to emphasise this in the revised text.

CHANGES:Added the following sentence to P15L6: "However, these dependencies should not be used to rectify the biased CF values, rather CF should be estimated with methods that do not result in biased CF values in the first place."

P16L33: Same remark as above: don't mix symbols and units in a formula.

CHANGES:Replaced the text in parentheses with "y=ax+b, where a=0.47 and b=-0.05 s for Siikaneva and a=0.44 and b=-0.07 s for Hyytiälä, where y is the time lag difference and x response time. Fits were based on orthogonal linear regression"

P16L34: the meaning of x and y is not fully clear to me. Could you express the relation between these variables and time constants presented above?

RESPONSE: x equals the estimated time constants for $H_2O$ in RH bins and y equals the mean time lag difference between $H_2O$ and $CO_2$ in RH bins. Here we assume that this time lag difference is caused solely by the low-pass filtering of $H_2O$ and hence is related to the tlpf as noted in the text.

CHANGES:No changes

P17L2, L5 and elsewhere: rather than referring to Section numbers, it would be more easy for the reader if you referred directly to the figures or tables presenting the results.

RESPONSE: We prefer referring to different sections rather than to individual figures or tables, since often the text is more meaningful point of reference than individual figures. However, we re-evaluated this and made the requested changes where applicable.

CHANGES:Referred to figures or tables instead of manuscript sections where applicable.

P17L7 and elsewhere: use a uniform notation to present the different methods ("Method X" is fine to me).

CHANGES:Fixed here and elsewhere in the text.

P17L9: one word is missing.

CHANGES:added "gave" between "methods" and "more"

P17L10: As Hyytialla is equipped with a LI7200 and Siikaneva with a LI7000, I would have expected the inverse: a lower tau value at Hyytialla. Could you comment ?

RESPONSE: Yes, this was also slightly puzzling to us. However, the stronger tau RH dependency at Hyytiälä than at Siikaneva was very evident in the data and hence likely not due to e.g. erroneous data processing. The difference might be related to rain caps: in Hyytiälä the used rain cap was the one recommended by ICOS, whereas in Siikaneva custom made rain cap was used. Aubinet et al. (2016) and Metzger et al. (2016) stressed the importance of rain caps on EC system high frequency response.

CHANGES:No changes

P17L12 and foll: This section (and the legend of Table 3) should be clarified: In the text, are you presenting difference between correction factors? between half hourly fluxes? between cumulated fluxes? On which period?  I finally supposed that you were comparing cumulated fluxes but this should be specified.

RESPONSE: We are comparing mean fluxes over the whole study period (given in Sect. 3.1). We are stating this by starting the sentence on P17L12 with "On average..." (comparison of relative errors of averages equals comparison of relative errors of cumulated fluxes).

CHANGES:We noted that the caption of Tables 2 and 3 might be a bit misleading and hence modified the first sentence for Table2 as "Relative difference between the mean sensible heat fluxes obtained with the different correction methods ((cor-ref)/ref*100 %) and the reference (unattenuated sensible heat flux)." and for Table 3 as "Relative difference between the mean $CO_2$ and $H_2O$ fluxes obtained with the different correction methods ((cor-ref)/ref*100 %)"

P17L12 and foll: I'm not convinced by the relevance of comparing relative differences on cumulated flux values. Indeed, relative values depend strongly on flux values (I suppose that

H2O flux values at night should be low and in these conditions larger relative errors do not mean much). In addition, the low error on cumulated values may also result from partial compensation of errors (for example during day and night). I have the same problem when I try to compare different correction methods on my data set and I'm not sure to have the best solution. I prefer comparing the fluxes by looking at the slope between the fluxes submitted to different corrections. Anyway, in view of the preceding remarks, I'm not sure that the fact that Method 3 gives the biggest difference at both sites (L16) is really relevant.

RESPONSE: As the correction methods tested here are all multiplicative, the comparison of mean fluxes (cumulated fluxes) is similar to the comparison of slopes between fluxes. In order to test this, we calculated the slopes between fluxes corrected with the four methods (Method 4 being the reference) using orthogonal linear regression and the obtained values are given in the table below. The values were transformed as (1-slope)*100 % in order to be comparable with the values given in Table 3 in the manuscript. In general, the values given below are close to the ones reported in Table 3 in the manuscript suggesting that comparison of relative differences between means and regression slopes gives a similar outcome. Biggest difference between the values presented below and in Table 3 were observed for Hyytiälä night time $H_2O$ fluxes. This was expected, since $H_2O$ fluxes were low during night time. Hence we opt to keep Table 3 in the manuscript as it is but add a note that the differences between the Methods were small.

| Site | Gas flux | Method 1 | | | Method 2 | | | Method 3 | | |
|---|---|---|---|---|---|---|---|---|---|---|
| | | All | Night | Day | All | Night | Day | All | Night | Day |
| Hyytiälä | $CO_2$ | -0.5 | -0.5 | -0.5 | +0.08 | +0.08 | +0.07 | -0.7 | -0.7 | -0.7 |
| Hyytiälä | $H_2O$ | +0.6 | +2.3 | +0.6 | -0.8 | -3.5 | -0.8 | -2.0 | -8.1 | -1.9 |
| Siikaneva | $CO_2$ | +0.4 | +0.2 | +0.4 | -0.3 | -0.2 | -0.3 | -2.4 | -1.1 | -2.5 |
| Siikaneva | $H_2O$ | +3.0 | +3.8 | +2.9 | -0.1 | -0.2 | -0.02 | -0.6 | -0.7 | -0.6 |

CHANGES:Added to the end of P18L1 "..., albeit the found differences between the Methods were generally small."

P22L1; I feel (of course!) concerned by the remark on our paper about the impact of dead volumes on the frequency response of gas sampling system. I could recognise that the fact that we didn't distinguish physical time lag from attenuation induced time lag led to cut off frequencies that are probably not really representative of the attenuation. However, the general decrease of the cut off frequency with increasing dead volumes (our Figures 5 and

6) and the need for reducing these volumes in the gas sampling system were important results that we showed in this paper, along those of Metzger et al. And this again reinforces my opinion that transfer functions based on observed cospectra and taking thus account of all attenuation processes affecting the system (even if in some cases we do not fully understand all of them) are to be preferred for routinely correcting measurements, as they provide more robust estimates of fluxes.

RESPONSE: This sentence was not meant to undermine the findings of these two important papers (Metzger et al., 2016; Aubinet et al., 2016), but rather to emphasise that the connection between time lags and response times has rarely been considered before (e.g. Ibrom et al. 2007). In addition, with this part of the text we tried to emphasise that new laboratory studies might be warranted or alternatively re-analysis of existing data from past laboratory tests. Note that this section of the manuscript was largely restructured due to Referee #2 comment.

CHANGES:No changes

**Referee #2: Johannes Laubach**

General comments

This manuscript is a valuable contribution aiming to improve the data correction methods for eddy-covariance (EC) measurements of trace gas fluxes. This is highly relevant because the EC method is used at hundreds of sites around the globe, often continuously for many years, to quantify the carbon exchange of vegetation, greenhouse gas source and sinks, and evaporation. The authors treat in detail, both theoretically and with experimental data, how the effects of high-frequency attenuation of gas measurements and of time lags between the gas and wind measurements influence and compound each other. With that, they clear up two points of debate and sometimes confusion (see below) and give guidance how EC gas flux computation algorithms should be organised. Given the widespread use of EC for gas flux measurements, this paper has potential for high citation count.

(1) The first point of the debate clarified here is that, with correctly determined physical lag time between wind and gas (or other scalar) signals, the transfer function for cospectral attenuation is equal to that for the scalar's power spectrum, not to its square root. Even though this has been shown in detail before (Horst 1997), it is worth stating again because the erroneous square root keeps appearing in recent eddy covariance methodology papers, such as Nemitz et al. (2018).

(2) The second point is novel and shows how things change if lag times between wind and scalar signals are determined using covariance maximisation (with uncorrected attenuated data). The covariance maximisation is a dubious yet widespread practice. It overestimates the lag time by including the phase shift effect of the low-pass filtering. The authors show that after covariance maximisation, the cospectral transfer function is no longer equal to that

derived from the power spectrum, and they derive a correction to compensate for this. This correction is the truly novel contribution of this paper.

Unfortunately, (2) gets a bit muddled up by the authors claiming that the cospectral transfer function after covariance maximisation is obtained approximately if the square root of the power spectrum's transfer function is used to correct the cospectrum, i.e. by reverting to the original misconception addressed in (1). In other words, one imperfect (or incorrect) processing step would be fortuitously compensated by another. I strongly suggest refraining from putting it this way, because a) in a mathematically exact sense, it is incorrect, and b) the approximation will quickly become inaccurate for lag times exceeding 1 second (outside the range tested in this paper). I explain these reservations further in the Specific Comments.

Overall, I think this manuscript is worthwhile publishing after revision (as detailed below) and with modified conclusions, along the following lines.

In my view, there should be a strong recommendation to discourage usage of the covariance maximisation method in the future. It is known (and nicely illustrated here) to be incorrect, by mixing two separate effects. In addition, it produces erratic unphysical results when fluxes are close to the detection limit (affecting typically most nighttime periods for $H_2O$ and the morning/evening transition periods for $CO_2$).

It is not that difficult to determine the physical lag time with other methods. Firstly, its expected value can be constrained by geometrical dimensions of tube and measurement cell, together with pressure and flow rate (which is known or even controlled for many gas analysers, and if not, a flow meter can be added). Lags due to clock mismatches or processing delays need to be included, too. Once the expected lag time is estimated in this way, it can be empirically confirmed (e.g. by popping a balloon filled with synthetic air next to the sonic anemometer and the air intake). Alternatively, the lag time can be determined AFTER obtaining the transfer function for the gas power spectrum and applying the corresponding low-pass filter to produce a degraded temperature spectrum: the correct lag time should be that which maximises the cross-correlation between the degraded temperature time series and the gas time series.

Determining the lag time with any of these methods, followed by the correct cospectral correction (1), should be the preferred procedure.

It is understandable that the authors wish to promote their novel correction, and I concede that it may be useful in some cases, especially where lag time determination with other methods is difficult or not possible any more (historical data). A modified conclusion to that effect would be acceptable.

However, the authors should not recommend reverting, after covariance maximisation, to using the square root of the power-spectral transfer function. That approach reminds me of the Copernican approach of retaining epicycles in the planetary orbits, in order to rescue the tenet of circular motion: a physically wrong and complicated correction of a calculation procedure that is necessary only because the original procedure is based on a flawed assumption. It may be reasonably accurate (as for the data used here) but nonetheless should be abandoned.

RESPONSE: We thank the referee for his positive feedback. In this comment the referee is arguing that we suggest using the square root of the power spectrum's transfer function (sqrt(H)) in spectral corrections. This was not our intention. Our intention was to show that sqrt(H) approximates the correct transfer function ($HH_p$) for cospectrum calculated from vertical wind speed (w) and attenuated scalar time series (c) after cross-covariance maximisation. We try to stress this more in the manuscript. However, it ought to be kept in mind that the description of eddy covariance (EC) system's high frequency response with Eq. (3) in the manuscript is also an approximation. In reality, several different phenomena impact the high frequency response (e.g. Moore, 1986) and lumping them all into one transfer function described with Eq. (3) in the manuscript is merely a practical approach also hampered with approximations. In this respect, the approximation of $HH_p$ with sqrt(H) is not necessarily crucial. Nevertheless, approximations should be avoided where possible.

We opt not to recommend the estimation of the physical lag time (tphys) via alternative means, as the referee suggests, however we will add a note about this in text. We agree with the referee that cross-covariance maximisation procedure is not ideal, but we disagree that estimation of tphys via alternative means is simple. There are several components which induce time lags between the recorded time series (signal travel time through sampling line, spatially displaced sampling locations, clock mismatches and processing delays etc.) and they all likely slightly vary in time. Hence all these processes should be continuously tracked in order to estimate tphys correctly. We argue that this procedure would easily result in biases of several 0.1 s in the tphys estimates. Remembering that the cross-covariance function between w and c has roughly an exponential shape, it can be concluded that biased tphys values would easily cause notable biases in flux estimates, especially if measurements are done close to the ground where turbulence time scales are small. The other approach suggested by the referee (i.e. estimation of tphys by using degraded temperature time series) is an interesting idea, however it can be conjectured to produce similar estimates as the Method 4 implemented in the manuscript since $H_p$ is the frequency domain counterpart for a constant time shift done in the time domain.

CHANGES: Added the following sentence to P20L20: "Note however, that sqrt(H) is an approximation, not an exact representation of the cospectral transfer function." Added the following sentences to the beginning of Sect. 2.2 (P5L2): "Ideally the time lag between $w'$ and $\hat{\chi}'$ would be estimated from the known dimensions of the gas sampling system and flow rate, in addition to other components causing time delay between signals. However, it is in practise difficult to keep track of all the components causing the time delay and hence a practical solution has been to estimate the time lag between $w'$ and $\hat{\chi}'$ via cross-covariance maximisation. This can be considered...".
* * *
Specific comments

Introduction and Theory as far as P 5 L 18 are very well written and I fully agree with the content. One minor point:

In P 3 L 22, the "^" notation is introduced prematurely and should be removed. Its introduction is repeated in P 4 L 15, and it is not used in any numbered equation until (9).

CHANGES:fixed

P 5 L 20-32

I contend that the statement "Co_m can be approximated by A_m" (L 20-21) is wrong. Covariance maximisation delivers a lag time t_used which differs from the true t_phys. The mathematical treatment to describe how the time series of w and X shifted by t_used combine with each other is completely analogous to Eqs (2) to (4) with phi_used replacing phi_phys. Eq (4) shows how there is always a frequency-dependent shifting of amplitude between Co_m and Q_m. In other words, it is not possible to make Q_m disappear for all f simultaneously. Hence, while Eq (10) is the correct result for A_m, it is incorrect to equate A_m with maximised Co_m. In fact, your statement on P 12 L 3-5 shows that you are aware of this, hence the text here should be revised to reflect that.

The paragraph starting in L 27 describes your procedure for estimating H_p. It does not stop with assuming H_p = 1/sqrt(H). This is actually the novel part of the Theory section, and it should be written out with numbered equations and clearly stated approximations, ending with the equation for H_p that you are using in practice (e.g. Fig. 4).

RESPONSE: We agree with the referee: it is not possible to make Q_m disappear with a constant time shift. However, please note that we are not saying that sqrt(H) exactly describes the attenuation of $Co_m$, but we are saying that after cross-covariance maximisation the attenuation of $Co_m$ can be **approximated** with sqrt(H). See Fig. 4 in the manuscript. There sqrt(H) does indeed **approximate** $HH_p$ (i.e. the correct transfer function) but is not exactly the same at all frequencies. Related to the paragraph starting in P5L27: there seems to be the same misunderstanding as in the case of Referee #1. The analyses and results presented here (P5L27-P6L2) and in Sect. 4.1 were not used in estimating tlpf for $H_p$ used in correction factor calculations. Rather these analyses merely reveal the interdependency between tau and tlpf from a more "theoretical" point-of-view. In practise tlpf was estimated as a fitting parameter when $HH_p$ was fitted to $H_{e,cs}$ in the Method 4, similarly as tau. See Table 1 in the manuscript. We try to clarify this in the text.

CHANGES:Added the following sentence to P6L2: "Note that the dependency between tlpf and tau derived here and in Sect. 4.1 was not used in the estimation of correction factors (see Sect. 3.2.1), but merely to analyse the factors controlling the dependency." Added the following sentence to P8L19: "In the Method 4, both parameters tau and tlpf, were estimated by fitting $HH_p$ to $H_{e,cs}$." Modified the sentence on P10L13-P11L2: "Nevertheless, these results indicate that, for a given site, tlpf can be approximated to be constant at a specific attenuation (tau) level and hence supporting the estimation of tlpf as a fitting parameter in the Method 4 when fitting $HH_p$ to $H_{e,cs}$ (see Sect. 3.2.1 and Table 1)."

P 6 top

In my view, the Theory section should not end here yet because you did actually take the analysis further, in Sections 4.1-4.2. It should at least be anticipated here that empirical analysis was used to get a better estimate of H_p.

CHANGES:Added the following sentence to P6L2: "These analyses are continued in Sect. 4.1."

P 8 L 15-18 Two comments: Firstly, "Method 1... is implemented e.g. in EddyPro after Hunt et al. (2016)": this is incorrect. The method CAN be implemented there, but does not have to. The user is free to choose which lag time determination is used, and Hunt et al. used a fixed lag time.

Secondly, if EddyPro is mentioned here, then it would be worthwhile clarifying that older versions implemented the "sqrt(H)" transfer function, while later versions implement the correct "H" transfer function. The change was made as a consequence of correspondence between Laubach (Hunt's co-author) and Fratini, as explained in a footnote of Hunt et al. (2016). So, for carrying out Method 3 with EddyPro, users would need to employ an older version.

RESPONSE: OK, thanks for the clarifications.

CHANGES:Added a note on P8L18 that older version of EddyPro followed Method 3. Added the following sentences to P8L20: "Note that also other techniques have been used to estimate the time lag (e.g. Hunt et al., 2016; Taipale et al., 2010) and the selection of the time lag estimation method may have an effect on the correct shape of the cospectral transfer function (see Sect. 2). For instance, the cross-covariance moving average method to estimate the time lag introduced in Taipale et al. (2010) should be related to Method 4 as it is also based on cross-covariance maximisation, whereas for the approach in Hunt et al. (2016) the correct cospectral transfer function is H (with the caveat that the physical time lag is estimated accurately)." This addition considers also a comment by the Referee #3.

P 8 L 19-20 "Throughout the study, cross-covariance maximisation was used...": that means that later when assessing CO2 and H2O flux data, no true reference (with correct lag time) is available - which is a pity!

"... as typically done...": I would be curious to know how widespread this practice really is (given the practical problems with small fluxes, when estimated lag times can be all over the place). Of course that is outside the scope of this paper. I'd just like to caution the authors that not every EC user does follow this practice, and as noted above, the EddyPro software offers alternative choices.

RESPONSE: Yes, it would be interesting to know how widespread the usage of cross-covariance maximisation is. As the referee points out, alternative approaches have also been used, especially in the case of small fluxes (e.g. Taipale et al., 2010; Langford et al., 2016; Nemitz et al., 2018; Kohonen et al., 2020). However, cross-covariance maximisation is a common approach in many measurement networks (e.g. ICOS and NEON) indicating its widespread usage.

CHANGES:No changes

P 9ff

I find Section 4.1 very hard to follow. First, C is frequency-dependent (Fig. 1 top), then it is set constant without clear motivation (Fig. 1 bottom), then wind-speed dependent (Fig. 2). Is it possible to rewrite this section to be less ad-hoc and with more rigour, and perhaps put relevant equations at the end of the Theory section (with a note that empirical coefficients will be determined in the Results)?

RESPONSE: We will try to improve this part of the manuscript.

CHANGES:Added the following sentence to P6L2: " The dependence between tlpf and tau can also be solved numerically based on $H_p \approx 1/\sqrt{H}$ and further assumption that tlpf is proportional to tau, i.e. tlpf=Ctau where C is a proportionality constant." We also partly rewrote the Sect. 4.1.

P 11 L 11 "as is typically done": cautionary note that this is an assumption about other users (as for P 8 already). Figure 3 is a very convincing illustration why the covariance-maximisation practice should be abandoned, and I'd love to see a statement to that effect!

RESPONSE: We acknowledge that other approaches are used as well, but argue that cross-covariance maximisation is the most widespread technique to take into account the time lags between signals.

CHANGES:No changes

P 12 L 2-5

"Q_m cannot be nullified": here the authors agree with my earlier comment. The sentence in L 3-5 should be moved to the Theory, below Eq (10).

"H H_p not exactly equal to sqrt(H)": in fact, the appearance of negative values means that the two expressions are fundamentally incompatible. As Fig. (4) shows, increasing the time lag has the effect of shifting the negative region towards lower frequencies, where it causes greater flux losses. The approximation as sqrt(H) then becomes inadequate quite quickly, with big effects in practice for "sticky" gases like H2O and NH3, where tau can easily exceed 1 second.

Since you actually have a method to compute H_p, with results shown in Fig. 4, I do not understand why you keep repeating the point about (inaccurate) resemblence to sqrt(H).

RESPONSE: We agree, the importance of $H_p$ increases with tau mostly because tlpf increases with tau. The inaccuracy of sqrt(H) did increase with tau and this was evaluated at moderate attenuation levels (see Method 3 for instance in Fig. 5a). We wanted to include the analysis on sqrt(H) in the manuscript since sqrt(H) has been widely used in correcting EC fluxes in the past and will, for the mentioned reasons, very likely be used in future, and hence this discussion helps in evaluating how much the fluxes corrected using sqrt(H) are possibly biased.

CHANGES:Moved sentences on P5L3-P5L5 to P5L25 with small changes. In response also to Referee #3 the following sentence was added to P12L3: "In particular, $HH_p$ shows negative values at high frequencies whereas sqrt(H) does not."

P 13 L 1 "somewhat underestimated CF": better quantify (about 5 %?)

CHANGES:Added "(approximately 4 % underestimation when tau=0.9 s)" on P13L2.

P 13 L 9-10, Fig. 5: I do not find Fig. 5c useful. The dependence of t_lpf on tau has been extensively covered in Section 4.1.

RESPONSE: We would like to keep Fig. 5c since it illustrates how well tlpf could be estimated when fitting $HH_p$ to $H_{e,cs}$. In addition, it illustrates the step-like behavior of tlpf which was caused by the discrete data time resolution (0.1 s).

CHANGES:No changes

P 14, Table 2: How come that Methods 2 and 3 have relative differences of order +/-1 % when Fig. 5a suggests correction factors about 5 % below the reference? Does that mean the largest fluxes systematically had the smallest corrections? Does the extended dataset behave differently to the smaller sample underlying Fig. 5? Some explanation of this is needed.

RESPONSE: We agree, this might require clarification. Note that the bias for Methods 2 and 3 in Fig. 5a when tau=0.7 s is approximately -2.4 % which is not too far from the corresponding values reported in Table 2 (-1.4 % and -1.5 % for Siikaneva). In general, if the unattenuated cospectra have large values (in a relative sense) at those frequency domains where the used cospectral transfer function does not describe well the true cospectral transfer function (here $HH_p$) then the spectral corrections will be biased. Hence the biases given in Table 2 should not be related to the flux magnitude, rather they should be connected to the accuracy of different methods in describing the cospectral transfer function at frequencies relevant for turbulent scalar transport. Hence they depend on the integral time scale of the transport as shown in Fig. 6.

CHANGES:Added the following sentence to P13L16: "In general, the values obtained with this extended dataset are in line with Fig. 5a for Siikaneva albeit slight differences can be noted."

P 15, Fig. 6: Why does Hyytiälä not show any negative values for Methods 2 and 3? Was the chosen tau range too small to simulate noticeable flux losses?

RESPONSE: Referee # 1 had a somewhat related question, please see our response to Referee #1. The referee is right: the flux losses were clearly smaller at Hyytiälä than at Siikaneva, however not negligible. At Hyytiälä, fluxes calculated with T filtered with tau=0.7 s were approximately 5 % lower (median; 25th and 75th percentiles for the relative difference were 3 % and 8 %) than unattenuated fluxes. This might partly explain the seemingly different behaviour in Figs. 6b and 6c. However, we speculate that the spectral characteristics of turbulent transport may differ between sites (different shape of cospectra, differences in quadrature spectra) since measurements in Hyytiälä were made above forest (roughness sublayer flow), whereas Siikaneva had low vegetation (boundary layer flow).

CHANGES:Added the following sentence to P14L9: "The different behaviour of the two sites in Figs. 6b and 6c may be related to differences in spectral characteristics of turbulent

P 17-18

Figs 7 and 8 are interesting. Unfortunately, as stated before, there is no true reference available because t_phys was not determined with any other method than covariance maximisation.

It may be useful in these figures (and perhaps already in Fig 4, too) to show some averaged or typical cospectrum in a second panel above or below the transfer functions, to give the reader an idea which cospectral regions were affected by the low-pass filtering. It seems that with the data used here it was a relatively minor part, hence flux losses were generally small. While this situation is highly desirable (meaning the experimental setup was near-optimal), it is not always achievable, and a different dataset with greater flux losses may lead to comparison statistics quite different to those in Tables 2 and 3. The last few lines on P 18 already hint towards that; perhaps make this discussion point a little bit stronger. This suggestion is based on my own experience at agricultural sites with mast height restrictions (2 m) to allow for irrigators moving overhead. There, particularly for H2O the flux corrections can be substantial, of order 30 to 50 %, in which case the correct shape of the transfer functions matters a lot more than in your datasets.

RESPONSE: Thanks, good suggestion. We will modify Figs. 4, 7 and 8 in the way that the referee suggests. We acknowledge that the flux losses due to low-pass filtering in our datasets were not large, but likely in the range of a typical EC system measuring gas fluxes.

CHANGES: Modified figures 4, 7 and 8 as the referee suggests. Also the corresponding figure captions were amended. Modified the sentence on P18L4 as "In these situations the correct shape of the cospectral transfer function matters a lot more than for the measurement setups included here and hence the variability between methods is expected to be larger".

P 20-22:

It would help the reader if the "Summary and Conclusions" section was shortened greatly, to "Conclusions" only, with clear recommendations on future data processing (less is more!).

Of the listed conclusions, the first should end with a recommendation to abandon the covariance maximisation method wherever possible (P 20 L 31).

The second (P 21 top) should be shortened to its first 3-and-a-half lines ("... caused by covariance maximisation"). The statement that "(H H_p) can be approximated with sqrt(H)" should be removed because it becomes highly questionable when the negative region of the transfer function reaches into lower, flux-carrying, frequencies. The content of L 5-9 is unnecessary repetition of points made in Section 4.4.

The third (P 21 L 10-17) does not lead to a clear recommendation (other than reinforcing that covariance maximisation is best avoided), so I suggest removing. The fourth (L 18-22) only repeats the second, remove. Instead, consider adding a recommendation to check whether

past Fluxnet datasets have been processed with a consistent method combination for determining lag time and transfer functions. Where an erroneous mix has been applied, the data should be reprocessed. A statement on the expected fractional changes from such reprocessing could be added (based on your Results, but somewhat speculatively with respect to other EC setups).

The last conclusion (L 23 to end) could be condensed and rephrased as an "outlook" towards which other aspects of cospectral corrections require further investigation.

RESPONSE: We agree, often less is more. Therefore, we followed the referees recommendation and significantly reduced the length of this section largely along the lines the referee suggested. However, we opt to keep the statement about sqrt(H) in the second point in the text. In addition, we disagree that the third point listed in the conclusions is not needed. In fact, we argue that this is one of the key messages in the manuscript. It is not only related to the cross-covariance maximisation as the referee suggests, but more generally to the fact that attenuated cospectrum is directly related to the flux underestimation (and hence the reason for the need of spectral corrections) and therefore the used spectral correction methods should be able to describe the attenuation of cospectrum accurately. For this reason Method 2 was found to be relatively accurate, despite it only approximating the correct cospectral transfer function.

CHANGES: Added the following sentence to the end of the first conclusion (P20L31): "Hence, investigation of other means for estimating the signal travel time might be warranted.". Shortened the second as the referee suggests. Added "roughly" between "can be" and "approximated" on P21L4. Combined the fourth point with the second and modified the text following the referee suggestions. The last conclusion was rewritten as "The theoretical framework proposed in this paper was able to describe the changes in $H_2O$ time lag as $H_2O$ response time increased with relative humidity. This suggests that the findings derived here with attenuated temperature time series are applicable also in real world situations for EC gas flux measurements. However, results for $CO_2$ deviated from the expectations made based on the theory. Reasons for this remained unclear, however low-pass filtering of inert gases (such as $CO_2$) in the gas sampling line may deviate from what was described here in such a way that the low-pass filter acting on $CO_2$ may not induce a phase shift. This should be investigated by using $CO_2$ flux data from an EC setup with pronounced attenuation of $CO_2$ fluctuations in the sampling line or alternatively in a controlled environment in a laboratory (Metzger et al., 2016; Aubinet et al., 2016)."
* * *
Minor technical comments

P 2 L 10 "trough" should be "through" (before "tubes")

CHANGES:fixed as suggested

P 2 L 15-19 (and possibly later): For easy reading, I suggest using one of the pairs "low-pass"/"high-pass" and "high-frequency/low-frequency", but not mixing the two.

CHANGES: Fixed in locations where relevant

P 6 L 11 & 12 It is usual practice to give town/city of the manufacturer, too, not just the country.

CHANGES:fixed as suggested

P 6 L 17 remove hyphen between "Sphagnum" and "species".

CHANGES:fixed as suggested

P 6 L 18 replace "with the height" with "with a height".

CHANGES:fixed as suggested

P 8 L 6 insert "to" between "prior" and "utilising".

CHANGES:fixed as suggested

P 18 Table 3 caption: "was used" should be "were used".

CHANGES:fixed as suggested

**Anonymous Referee #3**

General comments:

This manuscript discusses Eddy Covariance (EC) flux post-processing corrections. They present important clarifications to the theory and demonstrate the presence of a systematic bias in some standard post-processing software packages from time lags induced by low-pass filters in closed path EC systems when using the cross-covariance maximization technique to determine lag times. They also present a correction method to account for this effect and provide a thorough theoretical and empirical discussion of its implementation. This manuscript is well written, and the topic is well suited for AMT. The methods presented here are likely to prompt reprocessing of a number of historical flux data sets and the revision of some of recommended best practices for EC flux processing. I recommend publication after some revisions.

RESPONSE: Thanks for these positive comments.

CHANGES:No changes

Specific Comments:

P5 L27-L32: The equations presented here are quite important and should be broken out into numbered equations.

RESPONSE: While we agree that the derivations here are novel, we opt not to separate the equations into numbered equations. Note that these findings were not used in estimating tlpf when estimating the correction factors with the Method 4. By separating the equations here into numbered equations, it might leave an impression for a reader quickly skimming through the text that these equations were used in the flux corrections.

CHANGES:No changes

P6 L21: What is the calculated transit time through the sampling tube from the volume and flow rate (if that information is available).

RESPONSE: Unfortunately such information is not available and we add that the mole flow is the relevant entity here.

CHANGES:No changes

P9 L3-10: The limitations at higher attenuation levels needs further comment. Long responses times are common for studies of more reactive trace gases and it is not clear if this method should be considered in those cases. Some explicit guidelines for the reader about where this method breaks down would be useful.

RESPONSE: In hindsight, inclusion of even higher attenuation levels (tau > 0.7 s) in this study might have been warranted. In any case, the limitations of doing spectral corrections with $HH_p$ can be approximated to be the same as the limitations of empirical spectral corrections in general. That is, if the attenuation is strong at frequencies close to the cospectral peak frequency, then the normalisation required in Eq. (12) and (13) is uncertain and hence the estimated response times (and tlpf) are compromised.

CHANGES:Added the following sentence to P8L15: "All the four methods are subject to similar limitations, i.e. they are applicable only when the attenuation is not strong at frequencies close to the cospectral peak frequency."

P12 L1-3: This inequality seems important and draws into question the utility of approximating to sqrt(H) when you present a method to calculate H $H_p$

RESPONSE: Yes, Referee #2 also raised this issue. Please see the comments above.

CHANGES:Added the following sentence to P12L3: "In particular, $HH_p$ shows negative values at high frequencies whereas sqrt(H) does not."

P14 L8: Why does the bias in CF only increase linearly with τ/ITS at Siikaneva and not Hyytiälä?

RESPONSE: Both Referee #1 and #2 had somewhat related question, see above. In short, we do not have a direct answer, but we speculate that it might be related to differences in the spectral characteristics of the turbulent transport at these two sites (Hyytiälä=roughness

sublayer flow; Siikaneva=boundary layer flow).

CHANGES:Added the following sentence to P14L9: "The different behaviour of the two sites in Figs. 6b and 6c may be related to differences in spectral characteristics of turbulent transport at these two sites since measurements in Hyytiälä were made above forest (roughness sublayer flow) and in Siikaneva above short vegetation (boundary layer flow)."

P15 Fig6: It is interesting that the Hyytiälä data for unstable conditions shows a positive slope for Methods 2-4. Some discussion of this would be useful.

RESPONSE: Referee #1 had the same question, please see our response above. We speculate that it is related to the spectral characteristics of the turbulent transport at the site, namely to the quadrature spectrum (assumed zero throughout the study). Please note that this is already mentioned in the text (P15L1-P15L3).

CHANGES:No changes

P16 L4: Calculated t(phys) from tube volume and flow rate would be helpful here as well if that information is available.

RESPONSE: Yes, we agree. For Hyytiälä nominal flow rate was 10 l/min, tube length was 0.77 m and inner diameter 4 mm. Based on these values the signal travel time in the sampling line was approximately 0.06 s. This value is below the estimate based on cross-covariance maximisation. Similar calculations for Siikaneva were not possible as all the needed information was not available. Note also that unknown pressure and temperature distributions across the tube and longitudinally make the determination of the number moles air in the tube system and thus tphys uncertain.

CHANGES:Added the following sentence to P6L12: "A 0.77 m long tube (4 mm inner diameter) was used to sample air for the gas analyzer with a nominal flow rate of 10 l min$^{-1}$." Added the following sentence to P16L5: "For Hyytiälä the signal travel time calculated from the sampling tube dimensions and flow rate was 0.06 s."

P17 L17: Further comment on this discrepancy is needed. Is the implication that the response times were fast enough that the LPF induced lag time was minor compared to other potential factors?

RESPONSE: Yes, this is also our interpretation of the results. Note that this is already discussed in relation to the estimated response times (P16L6-P16L18).

CHANGES:Added the following sentence to P18L1: "Note that if tlpf ≈ 0, then both power spectra and cospectra are attenuated similarly (i.e. with H) yielding Methods 1, 2 and 4 to be similar."

P20 Summary and conclusions: It is addressed in the third point but I think it is warranted to be more direct in discouraging the use of Method 1 based on your clear demonstration of systematic biases using that method.

CHANGES:Added the following sentence to the end of Summary and conclusions section

(P22L4): "In particular, the usage of Method 1 is discouraged as it resulted in clearly biased flux values."

Misc: The use of cross-covariance moving average methods for determining lag times (as in Taipale et al. 2010) is becoming more common as an alternative to the cross-covariance maximisation. The attenuation induced lag time effect and your Method 4 correction should be equally valid when this method is used, but a brief comment would be useful.

RESPONSE: Thanks for the suggestion.

CHANGES: Added the following sentences to P8L20: "Note that also other techniques have been used to estimate the time lag (e.g. Hunt et al., 2016; Taipale et al., 2010) and the selection of the time lag estimation method may have an effect on the correct shape of the cospectral transfer function (see Sect. 2). For instance, the cross-covariance moving average method to estimate the time lag introduced in Taipale et al. (2010) should be related to Method 4 as it is also based on cross-covariance maximisation, whereas for the approach in Hunt et al. (2016) the correct cospectral transfer function is H (with the caveat that the physical time lag is estimated accurately)." This addition considers also a comment by the Referee #2.

Minor Technical Comments:

P8 L6: insert word "to" after "prior"

CHANGES:fixed as suggested

Figure 3. Recommend making the colors more distinguishable.

CHANGES:Modified the line colors

Figure 5 panel b. Why is Method 1 not plotted

RESPONSE: The response time is estimated in the Method 1 in the same way as in the Method 3 and therefore it was not plotted separately

CHANGES:Included the Method 1 to the legend in Fig. 5b. Note that a small coding error was observed during this process and it had a very minor effect on the correction factors shown for Method 4 in this particular figure. Other parts of the manuscript were not affected by this coding error.

P16 L10:  Wording of this sentence is unclear "can attain values only with the temporal resolution of the underlying data itself".

RESPONSE: As the data time resolution dictates the temporal resolution of the cross-covariance maximisation procedure, therefore the tlpf related to the cross-covariance maximisation can attain values only with the data temporal resolution.

CHANGES:Modified the sentence as "The low-pass filtering induced lag (tlpf) can attain values only with the temporal resolution of the data being processed (here 0.1 s)..."

References:

Aubinet, M., Joly, L., Loustau, D., De Ligne, A., Chopin, H., Cousin, J., Chauvin, N., Decarpenterie, T. and Gross, P.: Dimensioning IRGA gas sampling systems: laboratory and field experiments, Atmos. Meas. Tech., 9(3), 1361–1367, doi:10.5194/amt-9-1361-2016, 2016.

Finnigan, J.: Turbulence in Plant Canopies, Annu. Rev. Fluid Mech., 32(1), 519–571, doi:10.1146/annurev.fluid.32.1.519, 2000.

Hunt, J. E., Laubach, J., Barthel, M., Fraser, A. and Phillips, R. L.: Carbon budgets for an irrigated intensively grazed dairy pasture and an unirrigated winter-grazed pasture, Biogeosciences, 13(10), 2927–2944, doi:10.5194/bg-13-2927-2016, 2016.

Kohonen, K.-M., Kolari, P., Kooijmans, L. M. J., Chen, H., Seibt, U., Sun, W. and Mammarella, I.: Towards standardized processing of eddy covariance flux measurements of carbonyl sulfide, Atmos. Meas. Tech., 13(7), 3957–3975, doi:10.5194/amt-13-3957-2020, 2020.

Langford, B., Acton, W., Ammann, C., Valach, A. and Nemitz, E.: Eddy-covariance data with low signal-to-noise ratio: time-lag determination, uncertainties and limit of detection, Atmos. Meas. Tech., 8(10), 4197–4213, doi:10.5194/amt-8-4197-2015, 2015.

Metzger, S., Burba, G., Burns, S. P., Blanken, P. D., Li, J., Luo, H. and Zulueta, R. C.: Optimization of an enclosed gas analyzer sampling system for measuring eddy covariance fluxes of H2O and CO2, Atmos. Meas. Tech., 9(3), 1341–1359, doi:10.5194/amt-9-1341-2016, 2016.

Moore, C. J.: Frequency-Response Corrections for Eddy-Correlation Systems, Boundary-Layer Meteorol., 37(1–2), 17–35, 1986.

Nemitz, E., Mammarella, I., Ibrom, A., Aurela, M., Burba, G. G., Dengel, S., Gielen, B., Grelle, A., Heinesch, B., Herbst, M., Hörtnagl, L., Klemedtsson, L., Lindroth, A., Lohila, A., McDermitt, D. K., Meier, P., Merbold, L., Nelson, D., Nicolini, G., Nilsson, M. B., Peltola, O., Rinne, J. and Zahniser, M.: Standardisation of eddy-covariance flux measurements of methane and nitrous oxide, Int. Agrophysics, 32(4), doi:10.1515/intag-2017-0042, 2018.

Taipale, R., Ruuskanen, T. M. and Rinne, J.: Lag time determination in DEC measurements with PTR-MS, Atmos. Meas. Tech., 3(4), 853–862, doi:10.5194/amt-3-853-2010, 2010.

---

## Author Response (AR2)

Manuscript title: The high frequency response correction of eddy covariance fluxes. Part 2: the empirical approach and its interdependence with the time-lag estimation

Authors: Olli Peltola, Toprak Aslan, Andreas Ibrom, Eiko Nemitz, Üllar Rannik, and Ivan Mammarella

MS No.: amt-2020-479

Second review round

We thank the referee and the editor for their comments. Please find below point-by-point responses to the presented critique. Responses to the comments are in red and the corresponding changes to the manuscript are in blue.

**Associate Editor**

Comments to the Author:
Please consider the supplementary minor comments provided by Referee #2. These arguments are partly philosophical, but the referee does provide a sound mathematical argument for the pitfalls of covariance maximization.

In my own previous work, I have typically averaged all lag-covariance plots together to derive a single optimum lag for a species and then applied this across all EC calculations. This assumes that the lag is driven by invariant physical aspects of the experiment and avoids the bias issues mentioned by the reviewer; however, I expect that the "right" approach depends on the particular experiment.

I will leave it to the authors to decide whether or how to incorporate the Referee's comments. Science is a dialogue, and publications are an appropriate venue to carry on such debates.

RESPONSE: We thank the editor for his comments. We agree that further research is needed for finding a suitable approach for time lag estimation but argue that this is out of scope of the present article, whose focus is on spectral correction.

**Referee #2: Johannes Laubach**

General comments

The authors have certainly addressed all the specific criticisms and by that improved the manuscript. Also, I do not have the impression that there are any differences between authors and reviewers on the mathematics.

That said, I would still encourage the authors to make stronger recommendations for change

to "established" or "widespread" eddy flux processing procedures. I will elaborate more in the paragraphs below. I believe such strengthening of the messages would increase the impact of the paper and would be fruitful for the "flux community".

There seems to be some remaining disagreement between me and the authors on the relative merits of the cross-covariance maximisation method. While I would call it "dubious" and "flawed", the authors still describe it as a "practical solution". I do not doubt that the authors have understood my line of reasoning, so it comes down to a matter of opinion what degree of known imperfection is still considered acceptable. Ultimately it is the authors' call how to put this. I can only appeal to them to consider a more strongly critical wording, on the following grounds.

I have many years of experience with EC measurements with closed-path analysers near the ground, i.e. 2 m above agricultural surfaces. There, lag times are typically of order 0.3 to 3 s, and I have always found the cross-covariance maximisation method causing more trouble than a fixed lag time, even a somewhat inaccurate one, ever would. For all gases I've looked at ($H_2O$, $CO_2$, $CH_4$, $N_2O$, $NH_3$), covariance maximisation introduces frequent unrealistic, widely oscillating, lag time estimates that have nothing to do with instabilities in the sampling system. The reason for that is that correlation coefficients of w with a scalar variable are not huge to begin with (typically 0.2 to 0.4) and when one of the variables is attenuated, the peak of the correlation function becomes rather flat and thus hard to detect. Worse, because it is a "maximisation" procedure, the method selects against near-zero fluxes even when these are true. For example, measuring $N_2O$ fluxes after a long period without any nitrogen inputs to the soil should yield a single-peak flux histogram with a near-zero median. However, with the maximisation method, the histogram becomes double-peaked (one positive, one negative) because zero flux is actively avoided by the method. But this is not a reflection of separate source and sink processes, it is simply due to bias in the individual flux estimates (because random non-zero correlations, of either sign, get "detected" and locked on by the method). Gas fluxes that change sign twice daily, such as that of $CO_2$, are equally affected by this bias around the sign-change periods.

I would concede that cross-covariance maximisation can be useful as a diagnostic tool, because it allows early detection of clear trends (such as two clocks drifting apart). But still, actual flux values should be calculated with a lag time that is based on the technical flow parameters of the system and not automatically varying from run to run (apart from corrections for known clock drifts).

Hence, my opinion remains that the use of the cross-covariance maximisation method (in its present widespread form) should be discouraged, and if the present authors do not include a recommendation to this effect, they are missing a chance to influence the thinking of the "flux community" in this regard.

The following remarks are deliberately provocative. Please do not take them as personal criticism!

Can the authors perhaps ask themselves why they are reluctant to recommend abandoning the cross-covariance maximisation method? The fact that it is "widespread" is no good reason. If a method is known to be poor, it should be replaced with better ones. In this case, do the authors consider such change too laborious for users? Do they not wish to make this recommendation because, if followed widely, it would remove the need to use their here-presented correction procedure in the future? Do they fear conflict in the scientist networks they are involved in (in particular ICOS with its drive to standardise procedures)? Do they

worry about changes in the FLUXNET databases that could ensue from revised processing? I do not expect answers to these questions. They are merely intended to encourage the authors to be bolder in the paper's Discussion and Conclusions sections.

RESPONSE: We thank the referee for thoroughly reviewing our manuscript. We agree with the referee that ideally one would use the physical lag time to shift the time series and that cross-covariance maximization (CCM) is by no means a perfect approach (e.g. due to the issues raised by the reviewer (see also Langford et al., 2015) and due to the findings shown in this manuscript). However, we argue that accurate estimation of physical lag time (and its possible temporal changes) is not as simple as the referee suggest. Our analyses started from the assumption that CCM is used to shift the time series and we wanted to evaluate if this affects the frequency response corrections. The emphasis of our manuscript is on frequency response corrections, not on methods used to estimate the lag time. Hence, we argue that strong suggestions about abandoning CCM cannot be made based on our findings alone since there are other issues (e.g. the ones raised by the reviewer) related to the lag time determination that are not included in our manuscript. Such suggestions should be made in a separate paper focusing specifically on the lag time estimation methods.

Regarding the more "philosophical" comments from the reviewer: Scientists should always be ready to make a "paradigm shift" even thought it would mean a lot of additional work. That is how science moves forward. However, we argue that our study does not entail all the needed information for suggesting a paradigm shift for EC lag time estimation.

CHANGES: Added "(yet sometimes flawed (e.g. Langford et al.,2015))" after "practical" on P5L4. We modified the sentence "Hence, investigation of other means for estimating the signal travel time might be warranted" as "Hence, other means for estimating the physical time lag might be warranted, especially in the case of noisy measurements (Langford et al., 2015). Further research in this topic is required."
* * *
Specific comments (line numbers refer to the tracked-changes version)

P 5 L 23 and P 17 L 1

The authors have added information on flow and tube dimensions for Hyytiälä. The tube volume is so small that the total physical lag time is affected at a comparable rate by the time it takes to exchange the air in the volume of the Li-Cor measurement cell. In addition, there is a known processing lag if the LI-7550 Interface Unit is used. The three effects (tube, cell, processing) need to be added to estimate physical lag time.

RESPONSE: There's no discussion related to physical lag times on P5L23 and hence it is uncertain which part of the manuscript the reviewer is referring to.

CHANGES: Added that the processing lag needs to be considered to P17L1: "However, this calculation neglects the additional time lag caused by LI-7550 Interface Unit and hence can be assumed to slightly underestimate $t_{phys}$."

P 21 L 20-21

I believe the reader should be warned more clearly that the "approximation" is limited to the cases of relatively minor attenuation presented in this study. In particular, it should be noted that in cases where the phase effects cause sign reversals in the cospectrum, the sqrt(H) approach must fail.

CHANGES: Added the following sentence to P12L4: "Hence the approximation of $HH_p$ with sqrt(H) will likely fail under very strong attenuation."

P 21 L 32

In line with my General Comments, I find the inserted "Hence, investigating..." too weak. Why not say something like "Hence, other means for estimating the physical lag time should be used whenever possible." (Please do not use "signal travel time", that sounds like an electromagnetic phenomenon.)

CHANGES: We modified the sentence as: "Hence, other means for estimating the physical time lag might be warranted, especially in the case of noisy measurements (Langford et al., 2015). Further research on this topic is required."

References:

Langford, B., Acton, W., Ammann, C., Valach, A. and Nemitz, E.: Eddy-covariance data with low signal-to-noise ratio: time-lag determination, uncertainties and limit of detection, Atmos. Meas. Tech., 8(10), 4197–4213, doi:10.5194/amt-8-4197-2015, 2015.